# Double Wins: Boosting Accuracy and Efficiency of Graph Neural Networks by Reliable Knowledge Distillation

## Abstract

The recent breakthrough achieved by graph neural networks (GNNs) with few labeled data accelerates the pace of deploying GNNs on real-world applications. While several efforts have been made to scale GNNs training for large-scale graphs, GNNs still suffer from the scalability challenge of model inference, due to the graph dependency issue incurred by the message passing mechanism, therefore hindering its deployment in resource-constrained applications. A recent study (Zhang et al., 2022b) revealed that GNNs can be compressed to inference-friendly multi-layer perceptrons (MLPs), by training MLPs using the soft labels of labeled and unlabeled nodes from the teacher. However, blindly leveraging the soft labels of all unlabeled nodes may be suboptimal, since the teacher model would inevitably make wrong predictions. This intriguing observation motivates us to ask: *Is it possible to train a stronger MLP student by making better use of the unlabeled data?*

This paper studies cross-model knowledge distillation - from GNN teacher to MLP student in a semi-supervised setting, showing their strong promise in achieving a "sweet point" in co-optimizing model accuracy and efficiency. Our proposed solution, dubbed *Reliable Knowledge Distillation for MLP optimization* (**RKD-MLP**), is the first noise-aware knowledge distillation framework for GNNs distillation. Its core idea is to use a meta-policy to filter out those unreliable soft labels. To train the meta-policy, we design a reward-driven objective based on a meta-set and adopt policy gradient to optimize the expected reward. Then we apply the meta-policy to the unlabeled nodes and select the most reliable soft labels for distillation. Extensive experiments across various GNN backbones, on 7 small graphs and 2 large-scale datasets from the challenging Open Graph Benchmark, demonstrate the superiority of our proposal. Moreover, our RKD-MLP model shows good robustness w.r.t. graph topology and node feature noises. The code is available at `https://anonymous.4open.science/r/RKD-MLP-F2A6/`.

## 1 Introduction

Graph neural networks (GNNs), as the *de facto* neural architecture in graph representation learning (Zhou et al., 2020; Hamilton et al., 2017b), have achieved state-of-the-art results across a variety of applications, such as node classification (Kipf & Welling, 2016; Liu et al., 2020), graph classification (Ying et al., 2018; Gao & Ji, 2019), link prediction (Zhang & Chen, 2018; Zhang et al., 2021), and anomaly detection (Deng & Zhang, 2021; Chaudhary et al., 2019). Different from plain network embedding methods (Perozzi et al., 2014; Grover & Leskovec, 2016), GNNs rely on the convolution-like message propagation mechanism (Gilmer et al., 2017) to recursively aggregate messages from neighboring nodes, which are believed to improve model expressiveness and representation flexibility (Xu et al., 2018).

Despite the recent advances, GNNs are still facing several challenges during inference, especially when going deeper (Chen et al., 2020; 2021) and applying to large-scale graphs (Chiang et al., 2019; Zeng et al., 2019). The major reason (Abadal et al., 2021) is that the message propagation among neighbors from multi-hops always incurs heavy data dependency, causing substantially computational

costs and memory footprints. Some preliminary efforts attempt to fill the gap from different aspects. For example, (Zhou et al., 2021) proposes to accelerate inference via model pruning, and (Tailor et al., 2020) suggests to directly reduce computational costs by weight quantization. Although they can speed up GNNs to some extent, the improvements are rather limited, since the data dependency issue remains unresolved. Recently, GLNN (Zhang et al., 2022b) tries to tackle this issue by compressing GNNs to inference-friendly multi-layer perceptrons (MLPs) via knowledge distillation (KD). Similar to standard KD protocols (Hinton et al., 2015), GLNN trains the MLP student by using the soft labels from GNN teacher as guidance, and then deploys the distilled MLP student to conduct latency-constrained inference.

However, directly leveraging soft labels from the GNN teacher is suboptimal when the labeled nodes are scarce, a common scenario in graph-structured data (Kipf & Welling, 2016; Garcia & Bruna, 2017; Feng et al., 2020). This is mainly because a large portion of unlabeled nodes will be incorrectly predicted by GNNs due to its limited generalization ability. For instance, many GNN variants (Kipf & Welling, 2016; Veličković et al., 2017; Klicpera et al., 2018) can achieve 100% accuracy on the training set, yet their test accuracy is merely around 80% on Planetoid benchmarks. As a result, the soft labels of those wrongly predicted unlabeled nodes would introduce noises to the optimization landscape of the MLP student, leading to an obvious performance gap w.r.t. the GNN teacher (Zhang et al., 2022b).

To avoid the influence of mislabeled nodes, the common practice is to analyze their logit distributions from the teacher model (Kwon et al., 2020; Zhu et al., 2021a; Zhang et al., 2022a). For example, Zhang et al. (Zhang et al., 2022a) propose to assign larger weights to samples if their teacher predictions are close to one-hot labels. Zhu et al. (Zhu et al., 2021a) suggest filtering out data points if their teacher predictions mismatch with ground truth labels. Nevertheless, these methods cannot be applied in real-world graphs where node labels are expensive to access. Recently, Kwon et al. (Kwon et al., 2020) suggest discriminating samples based on entropy values, by assuming that teacher predictions with lower entropy are more reliable. However, we found that entropy values are ineffective to distinguish the correct and wrong decision boundaries of GNN models since they are often largely overlapped, as we show in Figure 1 (right panel). Therefore, it still remains an open challenge to effectively distill semi-supervised GNN models to light-weight MLP students.

**Present Work.** Motivated by this, we propose a novel KD framework – **RKD-MLP** to boost the MLP student via noise-aware distillation. It is noteworthy that while we focus on the MLP student for efficiency purposes, our solution is ready for other student types, such as GNNs (See Appendix F for more discussion). Specifically, RKD-MLP uses a meta-policy to filter out those unreliable soft labels by deciding whether each node should be used in distillation given its node representations. The student then only distills the soft labels of the nodes that are kept by the meta-policy. To train the meta-policy, we design a reward-driven objective based on a meta-set, where the meta-policy is rewarded for making correct filtering. The meta-policy is optimized with policy gradient to achieve the best expected reward and then be applied to unlabeled nodes. We iteratively update the meta-policy and the student model, achieving a win-win scenario: it substantially improves the performance of the vanilla MLP student by teaching it with reliable guidance while maintaining the inference efficiency of MLPs without increasing the model size.

- We provide the first comprehensive investigation of unlabeled nodes in GNNs distillation by demonstrating its validity in boosting the MLP student via providing effective pseudo labels, and perniciousness in degrading model performance via adding incorrect soft labels.

- Motivated by our analysis, we propose to use a meta-policy to filter out unreliable nodes whose soft labels are wrongly predicted by the GNN teacher, and introduce a bi-level optimization strategy to jointly train the meta-policy and the student model.

- Extensive experiments over a variety of GNN backbones on 7 small datasets and 2 challenging OGB benchmarks demonstrate the superiority of our proposal. Notably, our RKD-MLP outperforms the vanilla KD solution with up to 5.82% standard accuracy, while its inference is at least 100 times faster than conventional GNNs.

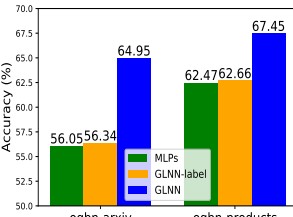 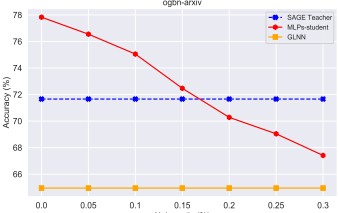 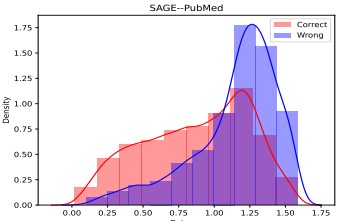

Figure 1: **Left:** The influence of unlabeled nodes on vanilla solution–GLNN (Zhang et al., 2022b). GLNN-label is a variant of GLNN by excluding unlabeled nodes. **Middle:** The impacts of wrongly predicted nodes on the MLP student under different noise ratios. **Right:** Entropy distributions of wrongly and correctly predicted nodes by GNN teacher. More curves are reported in Appendix E.

## 2 MOTIVATION

### 2.1 PRELIMINARIES

**Notations.** Let $\mathcal{G} = (\mathcal{V}, \mathcal{E})$ be a graph with $N$ nodes, where $\mathcal{V}$ and $\mathcal{E}$ stand for the node set and edge set, respectively. We use $\mathbf{X} \in \mathbb{R}^{N \times D}$ to denote node features, with row $\mathbf{x}_v$ being the $D$-dimensional feature vector of node $v \in \mathcal{V}$. We denote $\mathbf{Y} \in \mathbb{R}^{N \times C}$ as the label matrix with $C$ classes of interest, where $\mathbf{y}_v \in \mathbb{R}^C$ represents the one-hot label vector of node $v$. In semi-supervised learning, which is a common task in graph analysis, we have a small portion of nodes being labeled while the majority of the nodes are unlabeled. We mark labeled nodes with superscript$^L$, i.e., $\mathcal{V}^L$, $\mathbf{X}^L$, and $\mathbf{Y}^L$, and unlabeled nodes with superscript$^U$, i.e., $\mathcal{V}^U$, $\mathbf{X}^U$, and $\mathbf{Y}^U$.

**Graph Neural Networks.** GNNs are state-of-the-art neural network architectures for semi-supervised classification in graph analysis. Following the message passing mechanism (Gilmer et al., 2017), the core idea is to update an anchor node's representation by aggregating representations of itself and its neighbors. Formally, at the $k$-th layer, the updating rule is expressed as:

$$\mathbf{h}_v^{(k)} = \text{COM}(\mathbf{h}_v^{(k-1)}, \text{AGG}(\{\mathbf{h}_u^{(k-1)} : u \in \mathcal{N}_v\})), \tag{1}$$

where $\mathbf{h}_v^{(k)} \in \mathbb{R}^d$ denotes the representation of node $v$ at the $k$-th layer, and $\mathcal{N}_v$ is the neighbor set of $v$. The aggregation function AGG() aims to aggregate messages from neighbors via either fixed combinations determined by the graph topology (Kipf & Welling, 2016; Wu et al., 2019) or learnable attention weights (Veličković et al., 2017; Vaswani et al., 2017). The combination function COM() integrates the neighborhood information from the previous layer.

### 2.2 A CLOSER LOOK AT KNOWLEDGE DISTILLATION IN SEMI-SUPERVISED LEARNING

To bridge the gap between vanilla MLPs and more advanced GNNs for graph analysis, an intuitive solution is to conduct cross-model knowledge distillation. Formally, let $\mathbf{z}_v \in \mathbb{R}^C$ denote the soft labels of node $v$ predicted by a GNN teacher model, and $\hat{y}_v \in \mathbb{R}^C$ be the predictions of the MLP student model. The standard distillation process in (Zhang et al., 2022b) is expressed as:

$$\mathcal{L} = \lambda \sum_{v \in \mathcal{V}^L} \mathcal{L}_{CE}(\hat{\mathbf{y}}_v, \mathbf{y}_v) + (1 - \lambda) \sum_{v \in \mathcal{V}^L \cup \mathcal{V}^U} \mathcal{L}_{KL}(\hat{\mathbf{y}}_v, \mathbf{z}_v), \tag{2}$$

where $\mathcal{L}_{CE}$ is the standard cross-entropy loss on labeled nodes, while $\mathcal{L}_{KL}$ is **knowledge distillation loss**, i.e., the Kullback–Leibler divergence between the predictions of the MLPs student and GNNs teacher. $\lambda$ is a trade-off parameter. Note that, different from supervised learning, the distillation loss in Eq. 2 naturally includes two parts in the semi-supervised scenario: labeled node set $\mathcal{V}^L$ and unlabeled node set $\mathcal{V}^U$. This design choice is inspired by standard semi-supervised learning philosophy (Yang et al., 2021b), where unlabeled data is believed to be helpful in promoting model performance. According to our empirical results in Figure 1 (left), we observed that this tendency holds in KD. Without the soft labels of unlabeled nodes, the MLPs student can only perform comparably to the vanilla MLP baseline.

However, as aforementioned in the Introduction, we argue that blindly leveraging soft labels of all the nodes in Eq. 2 is suboptimal, since the soft labels from the teacher are noisy, especially for unlabeled nodes $\mathcal{V}^U$. Here, "noisy soft labels" refer to the soft labels of the nodes whose true labels mismatch

the predictions of the GNN teacher. To verify this point, we conduct preliminary experiments from the oracle perspective, by assuming that the ground truths of unlabeled nodes are known. Then, we manually control the ratio of noisy soft labels in the knowledge distillation loss. Figure 1 (middle) reports the results.

Observation: **The soft labels of incorrectly predicted nodes restrict the capacity of MLPs student; By reducing the noise ratios, a stronger MLPs student can be easily achieved.** As shown in the right panel of Figure 1, the MLP student's performance drops significantly as the noise ratio increases. If we can control the error ratio to some extent, e.g., 15%, the MLPs student can easily achieve comparable or even better results than GNNs teacher.

Nevertheless, it is a non-trivial task to effectively identify those wrongly predicted nodes from the correctly classified ones, following the standard entropy-based heuristic approach (Kwon et al., 2020). This is because the entropy distributions of the two groups are often largely overlapped in GNNs. For example, the entropy distributions of wrongly and correctly predicted nodes are generally overlapped with 40% areas on different GNNs models as shown in Figure 1 (middle) and Figure 9 (Appendix).

The above observations pose the challenge: *Can we filter out the noisy teacher guidance in an automatic fashion, such that a stronger MLP student can be achieved using reliable GNN knowledge?*

## 3    PROPOSED MODEL

In this part, we present a simple, generic, and effective KD framework to tackle the unreliable GNN guidance issues revealed in the Motivation section. Specifically, we first introduce the problem formulation in Section 3.1, and then elaborate on our proposal in Section 3.2.

### 3.1    PROBLEM FORMULATION

Given a graph $\mathcal{G} = \{\mathcal{V}, \mathcal{E}\}$, its feature matrix $\mathbf{X} \in \mathbb{R}^{N \times d}$, and label matrix $\mathbf{Y} \in \mathbb{R}^{N \times C}$. We use $\mathbf{Z} \in \mathbb{R}^{N \times C}$ to denote the soft label matrix produced by the teacher GNN model and $f_{\text{student}}$ to denote a student model parameterized by multi-layer perceptrons. In traditional knowledge distillation settings (Zhang et al., 2022b), the student model is optimized according to two soft-label sets: (1) the labeled set $\mathcal{R}^L = \{(\mathbf{x}_v, \mathbf{z}_v) | v \in \mathcal{V}^L\}$, and (2) the unlabeled set $\mathcal{R}^U = \{(\mathbf{x}_v, \mathbf{z}_v) | v \in \mathcal{V}^U\}$. Nevertheless, as discussed in Section 2, using the soft labels of all nodes in $\mathcal{V}$ would degrade the performance of student models, since many unlabeled nodes are incorrectly predicted by the teacher model, which introduces unreliable guidance.

To this end, we study the *reliable knowledge distillation* (RKD) problem. The core idea of RKD is to filter out the wrongly predicted nodes by GNN teacher and construct a reliable soft-label set (i.e., $\mathcal{R}_r$) for student training. Formally, $\mathcal{R}_r = \mathcal{R}_r^L \cup \mathcal{R}_r^U$ consists of two parts, where $\mathcal{R}_r^L$ (or $\mathcal{R}_r^U$) includes those labeled (or unlabeled) nodes that are correctly predicted by the GNNs teacher. In practice, we can directly obtain the soft-label set $\mathcal{R}_r^L$ from labeled nodes since we already have the ground truths. Specifically, given a labeled node $v$, if the prediction from the teacher matches the ground truth, then $v \in \mathcal{R}_r^L$; otherwise, $v \notin \mathcal{R}_r^L$. Therefore, the main challenge in RKD is how to determine the soft-label set $\mathcal{R}_r^U$ from unlabeled nodes, since no ground truth is available to check their validity.

### 3.2    THE PROPOSAL

We present RKD-MLP, a general reinforced framework for training student MLPs via reliable GNN distillation. The full framework is illustrated in Figure 2. The key idea is to learn a meta-policy network to determine the reliable soft label set (Section 3.2.1) and train the student MLP based on the reliable guidance (Section 3.2.2). After that, an unified framework is designed to train the meta-policy network and student model jointly (Section 3.2.3).

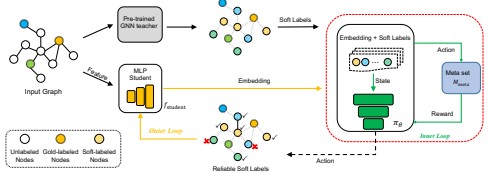

Figure 2: The RKD-MLP framework. Our meta-policy filters out noisy GNN teacher guidance, which is then used to train MLP student.

### 3.2.1 META-POLICY

To obtain the reliable soft label set $\mathcal{R}_r$, an intuitive solution is to utilize the uncertainty of teacher predictions (Kwon et al., 2020). For example, we can compute the entropy of all the nodes using their soft labels from the teacher GNNs, and then filter out those whose entropy values are higher than a pre-defined threshold. However, as shown in Figure 1 (Right) and Figure 9 (in Appendix), entropy can not well differentiate between the correct and incorrect nodes since they are largely overlapped. To overcome this limitation, we propose to develop a learning-based approach to automatically fit the complex decision boundary between them. Specifically, following (Zoph & Le, 2016; Gao et al., 2019; Lai et al., 2020), we assume that a meta-set with ground truth labels is available; in this work, we use the validation set as the meta-set. Then we propose to train a meta-policy with reinforcement learning (RL) to identify the reliable soft labels, where the meta-policy is updated based on the reward from the meta-set.

Formally, let $\mathcal{M}_{\mathrm{meta}} = \{(\mathbf{x}_v, \mathbf{z}_v, \mathbf{y}_v, \mathcal{I}_v)\}_{v=1}^m$ denote a meta-set with $m$ samples, where $\mathbf{z}_v$ is the teacher prediction and $\mathbf{y}_v$ denotes the ground truth. $\mathcal{I}_v = 1$ if the teacher model makes correct prediction; $\mathcal{I}_v = 0$ otherwise. We define the state, action and reward as follows. Let $\mathcal{S}$ be the state space; in this work, we use node representations (we will illustrate how to obtain these later) as the states, i.e., $\mathbf{x} \in \mathcal{S}$. Let $\mathcal{A} = \{0, 1\}$ be the action space, where 0 indicates that the soft label is unreliable, and 1 suggests that the soft label is reliable. Given a node $\mathbf{x}_v$, an agent takes action $a_v$ and receives a scalar reward $r_v$, where a positive reward $r_v = 1$ is given if the label is indeed reliable (i.e., correct teacher prediction) when $a_v = 1$ or indeed unreliable (i.e., incorrect teacher prediction) when $a_v = 0$, and $r_v = 0$ otherwise. Let $\pi : \mathcal{S} \rightarrow \mathcal{A}$ be a meta-policy that maps states to actions. With neural function approximators, we use $\pi_\theta$ to denote a parameterized meta-policy with parameters $\theta$ and $\pi_\theta(a|\mathbf{x}_v)$ to denote the probability of sampling $a$ at state $\mathbf{x}_v$. The objective is to train the meta-policy network $\pi_\theta$ such that it can maximize the expected reward:

$$\mathcal{J}_{\mathrm{meta}} = \mathbb{E}[r_v], \tag{3}$$

where node $v$ is any node from all the nodes in the graph. Following the policy gradient theorem (Williams, 1992), we can calculate the gradient of $\mathcal{J}$ w.r.t. $\theta$ as

$$\begin{aligned} \triangledown_\theta \mathcal{J}_{\mathrm{meta}} &= \triangledown_\theta \mathbb{E}[r_v] \\ &= \mathbb{E}[r_v \triangledown_\theta \log \pi_\theta(a_v|\mathbf{x}_v)], \end{aligned} \tag{4}$$

where $a_v$ is the currently selected action for node $v$. We approximate the above gradient with the samples in the meta-set $\mathcal{M}_{\mathrm{meta}}$:

$$\triangledown_\theta \mathcal{J}_{\mathrm{meta}} \approx \sum_{v \in \mathcal{M}_{\mathrm{meta}}} r_v \triangledown_\theta \left[ \log \pi_\theta(a_v|\mathbf{x}_v) \right], \tag{5}$$

where $r_v$ can be obtained based on the ground truths in the meta-set by regarding the reliable soft labels as the ones that the teacher model makes correct predictions. The update of Eq. 5 can be unstable due to the high variance of the gradients. Thus, we introduce a baseline (Sutton & Barto, 2018) for variance reduction. The final gradient can be written as

$$\triangledown_\theta \mathcal{J}_{\mathrm{meta}} \approx \sum_{v \in \mathcal{M}_{\mathrm{meta}}} (r_v - B) \triangledown_\theta \left[ \log \pi_\theta(a_v|\mathbf{x}_v) \right], \tag{6}$$

where the baseline $B = (\sum_{v \in \mathcal{M}_{\mathrm{meta}}} r_v)/m$ is the mean reward across the nodes in the meta-set, and $(r_v - B)$ can be interpreted as the advantage, i.e., the extent to which the current action is better than the average case. Then we use the meta-policy $\pi_\theta$ to predict the reliable soft labels for the unlabeled nodes. Specifically, the soft label of an unlabeled node $u$ is considered reliable if the predicted probability of $a = 1$ is larger than 0.5, i.e., $\mathcal{R}_r^U = \{(\mathbf{x}_u, \mathbf{z}_u)|\forall_u \in \mathcal{V}^U, \pi_\theta(a = 1|\mathbf{x}_u) > 0.5\}$.

▷ Rationale. Despite the simplicity of RL in Eq. 6, the reward-driven objective enables the meta-policy to reason about the reliability of the soft labels based on node features. Once trained on the meta-set, the meta-policy can transfer to the unlabeled nodes to take the most rewarding action (i.e., reliable or unreliable) for each node.

---

**Algorithm 1:** Alternating Gradient Descent for RKD-MLP

---

**Input:** Initial meta-policy network $\pi_\theta$ and initial MLPs student model $f_{\text{student}}$

**1 while** *not converge* **do**

**2**     1. Obtain the node embedding $\mathbf{h}$ of unlabeled nodes based on MLP student

**3**     2. Train the meta-policy network $\pi_\theta$ based on policy gradient in Eq. 6 and meta-set $\mathcal{M}_{\text{meta}}$

**4**     4. Fix meta-policy $\pi_\theta$ and update the student model $f_{\text{student}}$ based on reliable knowledge distillation loss in Eq. 7

**5 Return** The well trained MLP student model $f_{\text{student}}$.

---

### 3.2.2 STUDENT MODEL TRAINING WITH RELIABLE GUIDANCE

By querying the meta-policy, we can train the student MLPs with better guidance. Formally, we rewrite Eq. 2 as:

$$\mathcal{L} = \lambda \sum_{v \in \mathcal{V}^L} \mathcal{L}_{CE}(f_{\text{student}}(\mathbf{x}_v), \mathbf{y}_v) + (1 - \lambda) \sum_{v \in \mathcal{V}} \mathcal{I}_{\pi_\theta(a=1|\mathbf{x}_v)>0.5} \mathcal{L}_{KL}(f_{\text{student}}(\mathbf{x}_v), \mathbf{z}_v), \quad (7)$$

The key design of Eq. 7 is the second term, which only considers soft labels from the reliable set $\mathcal{R}_r$. One benefit of the above equation is that the soft labels are noise-less, so the MLPs student model will be trained with more reliable information from the teacher model.

### 3.2.3 THE UNIFIED TRAINING OBJECTIVE

Instead of training the meta-policy and student MLPs in a two-stage fashion, e.g., training the meta-policy first and then optimizing the student model, we propose to simultaneously train $f_{\text{meta}}$ and $f_{\text{student}}$ according to the following bi-level optimization framework:

$$\min_{f_{\text{student}}} \mathcal{L}(\mathbf{X}, \mathcal{Y}, f_{\text{meta}}^*) \quad \text{s.t.} \quad \pi_\theta^* := \arg\max_{\pi_\theta} \mathcal{J}_{\text{meta}}(\mathcal{M}_{\text{meta}}, f_{\text{student}}^*) \quad (8)$$

The outer objective $\mathcal{L}$ is defined in Eq. 7, which requires the meta-policy $\pi_\theta$ to select reliable soft-label predictions from the GNN teacher. The inner objective $\mathcal{J}_{meta}$ is defined in Eq. 6, and it takes node representations from the MLP student and soft label vectors from the GNN teacher as input. It is worth noting that the design of $\pi_\theta$ can take other node embeddings as input such as the raw features or hidden embedding from the GNN teacher. However, we find that using $\mathbf{h}_u$ as state representation is beneficial since jointly training the policy network and MLP student could reinforce their reciprocal effects. This is because learning a better MLP student requires $\pi_\theta$ to generate a more reliable soft label set while training a high-qualified policy needs more informative node embeddings as input. From Table 1 and 2, we can see that the MLP student performs better than the corresponding GNN teacher when it converges. Thus, it is reasonable to conjecture that hidden representations of the MLP student are more informative. Following the common practice (Liu et al., 2018; You et al., 2021), we adopt the Alternating Gradient Descent (AGD) algorithm to optimize Eq. 8, by iteratively updating the outer and inner optimization objectives, as outlined in Algorithm 1.

## 4 EXPERIMENTS

In this section, extensive experiments are reported to explore the following research questions. **RQ1:** How effective is RKD-MLP compared with state-of-the-art baselines in transductive and inductive settings? **RQ2:** Can RKD-MLP scale up to large-scale graphs? **RQ3:** What are the impacts of noisy node features or topology structures on RKD-MLP? **RQ4:** How effective is our meta-policy in identifying reliable teacher guidance? **RQ5:** How does each component of RKD-MLP contribute to the performance? **RQ6:** How efficient is RKD-MLP compared with other acceleration methods?

### 4.1 EXPERIMENTAL SETUP

**Benchmark Datasets.** For comprehensive comparison, we use seven popular semi-supervised classification datasets with various scales and types, including **Cora**, **CiteSeer**, and **PubMed** (Sen et al., 2008), **WikiCS**, Amazon-Computers (**Compute**), Amazon-Photo (**Photo**), Coauthor-CS

Table 1: Node classification accuracy on commonly used graph datasets in transductive learning. *Improv.* indicates our proposal outperforms GLNN baseline. "-" indicates the result for the teacher.

| Teacher | Student | Cora | CiteSeer | PubMed | WikiCS | Compute | Photo | CS |
|---------|---------|------|----------|--------|--------|---------|-------|-----|
| MLPs | - | $58.04 \pm 0.75$ | $59.22 \pm 1.31$ | $70.54 \pm 0.77$ | $63.73 \pm 1.51$ | $67.80 \pm 1.06$ | $78.77 \pm 1.74$ | $84.80 \pm 0.59$ |
| SAGE | - | $79.70 \pm 0.52$ | $68.59 \pm 0.27$ | $76.55 \pm 0.29$ | $65.59 \pm 0.88$ | $82.97 \pm 2.16$ | $90.90 \pm 0.84$ | $90.56 \pm 0.38$ |
| | SW | $48.88 \pm 6.81$ | $54.47 \pm 9.39$ | $76.48 \pm 0.59$ | $58.77 \pm 2.28$ | $56.30 \pm 3.31$ | $63.15 \pm 4.85$ | $37.18 \pm 1.22$ |
| | Entropy | $80.73 \pm 0.76$ | $69.48 \pm 0.52$ | $78.48 \pm 0.29$ | $69.02 \pm 1.41$ | $83.53 \pm 0.65$ | $92.42 \pm 0.74$ | $91.79 \pm 0.39$ |
| | Cluster | $74.60 \pm 0.35$ | $70.23 \pm 0.55$ | $80.66 \pm 0.07$ | $67.05 \pm 0.67$ | $81.48 \pm 0.95$ | $87.84 \pm 0.35$ | $91.54 \pm 0.15$ |
| | GLNN | $80.00 \pm 0.52$ | $68.95 \pm 0.70$ | $76.97 \pm 0.28$ | $67.82 \pm 1.36$ | $83.04 \pm 1.70$ | $92.02 \pm 1.15$ | $90.95 \pm 0.51$ |
| | RKD-MLP | $81.52 \pm 0.66$ | $70.23 \pm 0.48$ | $80.97 \pm 0.20$ | $71.77 \pm 0.64$ | $84.23 \pm 0.53$ | $93.78 \pm 0.46$ | $92.58 \pm 0.30$ |
| | *Improv.* | +1.90% | +1.86% | +5.20% | +5.82% | +1.43% | +1.91% | +1.79% |

Table 2: Node classification accuracy on commonly used graph datasets in inductive learning. *Improv.* indicates our proposal outperforms GLNN baseline. "-" indicates the result for the teacher.

| Teacher | Student | Cora | CiteSeer | PubMed | WikiCS | Compute | Photo | CS |
|---------|---------|------|----------|--------|--------|---------|-------|-----|
| GCN | - | $80.29 \pm 0.19$ | $72.64 \pm 0.45$ | $78.88 \pm 0.21$ | $66.13 \pm 0.15$ | $80.33 \pm 0.57$ | $86.34 \pm 0.57$ | $89.02 \pm 0.48$ |
| | SW | $47.12 \pm 4.07$ | $59.08 \pm 6.00$ | $76.88 \pm 0.81$ | $54.40 \pm 2.23$ | $48.58 \pm 10.25$ | $62.39 \pm 6.65$ | $35.81 \pm 0.54$ |
| | Entropy | $80.87 \pm 1.26$ | $73.53 \pm 0.77$ | $79.28 \pm 0.84$ | $68.73 \pm 0.70$ | $82.30 \pm 1.08$ | $89.00 \pm 0.83$ | $91.59 \pm 0.27$ |
| | Cluster | $72.30 \pm 0.52$ | $75.12 \pm 0.63$ | $80.22 \pm 0.32$ | $65.63 \pm 0.55$ | $81.62 \pm 0.68$ | $87.30 \pm 0.74$ | $91.34 \pm 0.17$ |
| | GLNN | $79.23 \pm 1.51$ | $73.42 \pm 0.41$ | $79.36 \pm 0.32$ | $67.74 \pm 0.49$ | $80.56 \pm 0.73$ | $87.40 \pm 0.57$ | $90.64 \pm 0.23$ |
| | RKD-MLP | $82.56 \pm 0.74$ | $75.22 \pm 0.71$ | $81.00 \pm 0.50$ | $69.88 \pm 0.38$ | $82.47 \pm 0.49$ | $90.36 \pm 1.22$ | $92.38 \pm 0.31$ |
| | *Improv.* | +4.20% | +2.45% | +2.07% | +3.16% | +2.37% | +3.38% | +1.92% |

(**CS**) (Shchur et al., 2018). For experiments on large-scale graphs, we use two Open Graph Benchmark datasets (Hu et al., 2020): **ogbn-arxiv** and **ogbn-products**. More detailed discussions about the datasets and their statistics are included in Appendix A.

**Teacher GNNs.** For a thorough comparison, we consider five promising GNNs architectures as teacher models in our knowledge distillation framework: GraphSAGE (Hamilton et al., 2017a) (**SAGE**), **GCN** (Kipf & Welling, 2016), **APPNP** (Klicpera et al., 2018), **GAT** (Veličković et al., 2017), and **SGC** (Wu et al., 2019). For extremely large-scale datasets such as ogbn-product, we consider two scalable teacher GNNs: **ClusterGCN** (Chiang et al., 2019) and **GraphSAINT** (Zeng et al., 2019). The detailed training settings of these teacher models are listed in Appendix B.1.

**Student Competitors.** In addition to the GNN teachers, we also include two types of student baselines for comparison. First, we consider three heuristic-based approaches: **Cluster**, **Entropy**, and sample re-weighting (**SW**), which construct reliable soft-label set via clustering, relative prediction rankings, and sample re-weighting, respectively (See Appendix B.2 for more details). Second, we include two MLPs based related work: vanilla **MLPs** and **GLNN** (Zhang et al., 2022b).

**Transductive vs. Inductive.** Follow previous studies (Zhang et al., 2022b), we evaluate our model under two node classification settings: transductive and inductive. The main difference between them is whether to use the test data for training or not. For the inductive setting, the test nodes as well as their edge links will not be used. The experiment details of the two settings are in Appendix B.3.

#### 4.1.1 IMPLEMENTATION DETAILS

We build our model based on Pytorch and PyG library (Fey & Lenssen, 2019). For GNN teachers, following common practice in (Zhu et al., 2021b; Hu et al., 2020; 2021a), we employ a three-layer GNN encoder with dimension $d = 256$ for OGB benchmarks (ogbn-arxiv, and ogbn-products), while a two-layer GNN encoder with dimension $d = 128$ for other datasets. For MLP students, following (Zhang et al., 2022b), we set the number of layers and the hidden dimension of each layer to be the same as the teacher GNN. We set $\lambda = 0$ if not specified, since we empirically found that the proposed model is robust to $\lambda$ as shown in Figure 10 of Appendix. All the experiments are run 5 times on GeForce RTX 2080 Ti GPUs, and we report the mean and the standard deviation. More detailed configurations for different GNN teachers and MLP students are provided in Appendix B.4.

### 4.2 RESULTS AND ANALYSIS

**How effective is RKD-MLP against other baselines on small datasets? (RQ1).** Table 1& 2, and Table 6& 7 (in Appendix) report the results of our RKD-MLP with heuristic and MLPs based baselines. We make three major observations. **First**, compared with vanilla MLPs and intuitive

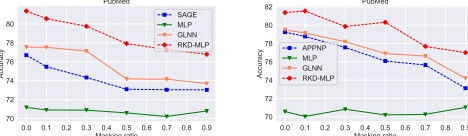 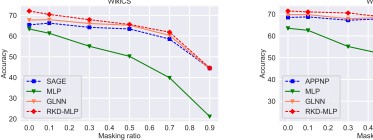

Figure 4: Accuracy results of RKD-MLP and other baselines w.r.t. noise graph topology.

Figure 5: Accuracy results of RKD-MLP and other baselines w.r.t. noise node features.

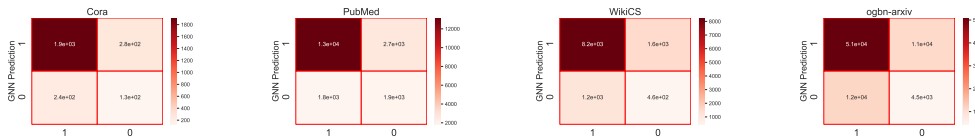

Figure 6: Confusion matrix of RKD-MLP on unlabeled nodes using GNN prediction as ground truths. The x-axis is the prediction of our proposal, and y-axis denotes if GNN teacher makes the right prediction. 1 means make the right prediction; otherwise 0.

KD method - GLNN, RKD-MLP performs significantly better than them in all cases. Specifically, RKD-MLP improves GLNN by up to 5.82% in the transductive setting (See Table 1). **Second**, RKD-MLP also outperforms three heuristic solutions (SW, Entropy, and Cluster) in general. The possible explanation is that our meta-policy is trained end-to-end with the MLP student, so that they can reinforce their reciprocal effects. **Third**, compared with 5 GNN teachers, our proposal consistently achieves better results across different benchmark datasets and two evaluation scenarios. Another interesting result is that the two heuristic methods (Entropy and Cluster) generally perform on par with or even better than GLNN across two settings. These results shed light on our motivation to study reliable knowledge distillation for MLP student training.

**How does RKD-MLP perform on large-scale graphs? (RQ2)**. Figure 3 and Figure 7 in Appendix summarize the results on two challenging large-scale graphs like ogbn-arixv and ogbn-products, from which we derive two insights. **First**, RKD-MLP is scalable and can achieve much higher results than vanilla KD method. As shown in Figure 3, RKD-MLP improves GLNN 8.5% and 6.3% on ogbn-arxiv and ogbn-products, respectively. **Second**, unlike small datasets, it is hard to train the MLP student on large graphs due to soft-label noises. For instance, GLNN can achieve comparable results with GNN teacher on small datasets (See Table 1), but performs significantly worse on large graphs (See Figure 3& 7). By avoiding unreliable guidance, our RKD-MLP can easily outperform the GNN teacher on small datasets, and bridge the gap between GLNN and the GNN teacher on large graphs.

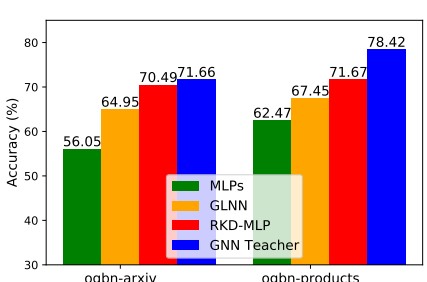

Figure 3: Accuracy results of RKD-MLP on large-scale graphs. **Left**: GraphSAGE teacher. **Right**: clustergnn teacher.

**How robust is RKD-MLP w.r.t. feature or topology noises? (RQ3)**. Figure 4& 5 report the results of RKD-MLP on two types of noise scenarios (Detailed experimental setups are provided on Appendix D.1). In general, we observe that our proposal performs consistently better than other baselines, and is more robust on topology noise compared with feature noise. For example, the performance gap between RKD-MLP and the second best baseline on incomplete graph structure (left two panels) is higher than that on noise feature (See Figure 4). We contribute this robustness gain to the proposed meta-policy, since it can filter out noisy teacher guidance.

**How effective is RKD-MLP in eliminating noisy guidance? (RQ4)**. We summarize the confusion matrix produced by our meta-policy to study its effectiveness. As shown in Figure 6, our proposal can effectively reduce the noise degree to some extent across different datasets (truth positive or negative samples). For instance, RKD-MLP reduces the relative noisy ratio from 23.26% to 17.20% on PubMed, and 29.42% to 18.03% on ogbn-arxiv.

**Ablation Study (RQ5)**. We study the importance of joint optimization and random selection on RKD-MLP with two ablations: RKD-MLP-iso and RKD-MLP-rand. RKD-MLP-iso is obtained

Table 3: Ablation study of RKD-MLP. clustergcn teacher for products while SAGE for others.

| | Cora | CiteSeer | PubMed | WikiCS | Computer | Photo | CS | ogbn-arxiv | ogbn-products |
|---|---|---|---|---|---|---|---|---|---|
| RKD-MLP-iso | 79.66 | 69.10 | 77.65 | 68.65 | 82.87 | 91.57 | 90.76 | 65.86 | 67.86 |
| RKD-MLP-rand | 79.22 | 68.25 | 76.32 | 67.35 | 82.28 | 91.16 | 90.03 | 64.52 | 66.88 |
| RKD-MLP | **81.59** | **70.46** | **81.07** | **71.77** | **84.23** | **93.78** | **92.58** | **70.49** | **71.67** |

by separating the training of meta-policy and the MLP student. RKD-MLP-rand is obtained by replacing the meta-policy with random selection. We made two observations from Table 3. **First**, jointly optimizing meta-policy and MLP student can reinforce their reciprocal effects, since RKD-MLP outperforms RKD-MLP-iso in all cases by a great margin. **Second**, random selection fails to distinguish the decision boundary between correctly predicted samples and incorrectly classified samples by the GNN teacher, so it performs the worst.

**Efficiency Analysis (RQ6)**. We compare the inference efficiency of RKD-MLP against state-of-the-art acceleration methods based on sampling, pruning and quantization strategies in Appendix C. From the results in Table 5 in Appendix, we observe that our distilled MLP student runs significantly faster than all other baselines. Given the high accuracy results of RKD-MLP on Table 1 and 2, our model is more desired to be deployed on resource-sensitive applications.

## 5 RELATED WORK

**GNN Acceleration.** The existing efforts for GNN speedup can be mainly grouped into two categories: scalable training and inference acceleration. **1)** Scalable training aims to scale GNNs to large-scale graphs with millions or even billions of nodes. Typical examples including sampling (Hamilton et al., 2017a; Chen et al., 2018; Zeng et al., 2019), clustering (Chiang et al., 2019), and decoupling (Wu et al., 2019; Rossi et al., 2020; Sun et al., 2021) based methods. Unfortunately, these methods suffer from inference challenges, since their message propagation process (Gilmer et al., 2017) is computationally expensive. **2)** Inference acceleration focuses on accelerating the inference latency of GNNs, in order to extend their applicability in resource-constrained applications. Several initial attempts based on pruning (Zhou et al., 2021; Chen et al., 2021; Sui et al., 2021) and quantization (Zhao et al., 2020; Liu et al., 2021) have been developed, but their improvements are limited, cause they still rely on message propagation for embedding. In contrast, our RKD-MLP results in a pure MLP, which naturally runs significantly faster. Graph-MLP (Hu et al., 2021b) also tries to learn an MLP for node classification, but it only considers transductive setting but not the more practical inductive setting.

**GNN distillation.** Knowledge distillation KD has been applied to compress GNNs recently. However, the majority of them are designed to distill large GNNs to smaller GNNs (Yang et al., 2020; Yan et al., 2020; Deng & Zhang, 2021; Xu et al., 2020) or label propagation student (Yang et al., 2021a). Since message propagation is also needed, they face the same inference issue as standard GNNs. One close work to us is GLNN (Zhang et al., 2022b), which accelerates GNN inference by distilling it to an MLP student. However,s it only considers utilizing knowledge from a GNN teacher blindly, which is suboptimal in practical classification situations when labeled nodes are scarce, causing the teacher model will inevitably deliver wrong predictions. In contrast, our proposal proposes to say no to noisy teacher knowledge and train an MLP student using those reliable soft labels.

## 6 CONCLUSIONS AND FURTHER WORK

In this paper, we study how to distill GNN knowledge to an MLP student when labeled nodes are scarce. We empirically found that the utilization of unlabeled nodes is the key to training the MLP student successfully. Meanwhile, the noise in soft labels of unlabeled nodes is the major hurdle to preventing the student model from growing stronger. To this end, we present RKD-MLP, an effective distillation framework, which offers reliable teacher guidance for the MLP student by filtering out incorrectly predicted unlabeled nodes automatically. Extensive experiments on open-sourced small and large benchmarks demonstrate the effectiveness&efficiency of RKD-MLP over existing algorithms. Moreover, RKD-MLP shows promising robustness w.r.t. incomplete graph topology and noise node feature scenarios, making it a desirable choice in resource-constrained applications. In the future, we will explore how to employ multiple teachers to further improve the performance of RKD-MLP, since different GNN variants may have different prediction capacities in practice.

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

Table 4: Dataset Statistics.

| Data | # Nodes | # Edges | # Features | # Classes |
|---|---|---|---|---|
| Cora | $2,708$ | $5,429$ | $1,433$ | 7 |
| CiteSeer | $3,312$ | $4,660$ | $3,703$ | 6 |
| PubMed | $19,717$ | $44,338$ | $500$ | 3 |
| Wiki-CS | $11,701$ | $216,123$ | $300$ | 10 |
| Amazon-Computers | $13,752$ | $245,861$ | $767$ | 10 |
| Amazon-Photo | $7,650$ | $119,081$ | $745$ | 8 |
| Coauthor-CS | $18,333$ | $81,894$ | $6,805$ | 15 |
| ogbn-arxiv | $169,343$ | $1,166,243$ | $128$ | 40 |
| ogbn-products | $2,449,029$ | $61,859,140$ | $100$ | 47 |

## A  DATASETS

In this section, we introduce the details of the datasets as bellow.

- **Cora, CiteSeer**, and **PubMed**: They are three widely used citation network datasets. Nodes represent documents and edges denote citation links. Each node has a sparse bag-of-the-words feature vectors. Labels are defined as the academic topics. We use them for both link prediction and node classification tasks.

- **ogbn-arxiv**: This is a large-scale paper citation network. Each node is an arxiv paper and the edge indicates the citation connection between two papers. Every node is associated with a 128-dimensional feature vector obtained by averaging the embeddings of words in its title and abstract. The task is to predict the 40 subject areas of arxiv CS papers.

- **Wiki-CS**: It is a reference network constructed based on Wikipedia. The nodes correspond to articles about computer science and edges are hyperlinks between the articles. Nodes are labeled with ten classes each representing a branch of the field. Node features are calculated as the average of pretrained GloVe word embeddings of words in each article.

- **ogbn-products**: This is an undirected and unweighted graph, representing an Amazon product co-purchasing network. Nodes represent products sold in Amazon, and edges between two products indicate that the products are purchased together. Node features are dimensionality-reduced bag-of-words of the product descriptions. The task is to predict the category of a product in a multi-class classification setup, where the 47 top-level categories are used for target labels.

- **Amazon-Computers** and **Amazon-Photo**: They are two networks of co-purchase relationships constructed from Amazon. Nodes indicate goods and edges represent the co-purchase relationships of two products. Each node has a sparse bag-of-words feature encoding products reviews and is labeled with its category. They are widely used for node classification task. Nodes represent authors and edges indicate co-authorship relationships. Each node has a sparse bag-of-words feature based on paper keywords of the author. The task is to predict the most active research field of authors.

- **Coauthor-CS**: This is an academic network, which represents co-authorship graphs based on the Microsoft Academic Graph from the KDD Cup 2016 challenge.

We follow standard semi-supervised setting to split the data. Specifically, for Cora, CiteSeer, and PubMed datasets, we use the GCN (Kipf & Welling, 2016) splitting strategy with the default random seed; for WikiCS, Amazon-Computers, Amazon-Photo, Coauthor-CS, we use the CPF (Yang et al., 2021a) splitting strategy and each random seed corresponds to a different split. For the OGB datasets, we follow the OGB official splits based on time and popularity for Arxiv and Products respectively.

# B  EXPERIMENTAL SETUP

## B.1  TEACHER ARCHITECTURES

The hyperparameters of GNN models on each dataset are taken from the best hyperparameters provided by the CPF (Yang et al., 2021a) paper and the OGB official examples. Table 8 and 9 summarize the hyperparameter configurations for 7 GNN models: GCN, GraphSAGE, APPNP, GAT, SGC, ClusterGCN, and GraphSAINT.

## B.2  STUDENT BASELINE

To have a fair comparison, for all two MLP student models (RKD-MLP and GLNN) and the vanilla MLP baseline, we set the number of layers and the hidden dimension of each layer to be the same as the teacher GNN, so their total number of parameters stays the same as the teacher GNN.

In addition to the MLPs based baselines, we also introduce several **heuristic** based competitors, which construct the reliable soft label set via rules of thumb.

- **Cluster.** It is a heuristic approach using clustering algorithm. The core idea is to construct reliable soft label set from unlabeled nodes by verify their cluster assignments. Specifically, Cluster first builds $C$ centroids, one for each label class, and then assigns each unlabeled nodes to one of $C$ clusters according to their similarities in the feature space. Finally, an unlabeled label will be marked as reliable if its prediction from the teacher matches the assigned cluster number.
- **Entropy.** It constructs the reliable soft label set using the relative ranking of unlabeled nodes based on their entropy scores from the teacher model. In our experiments, we tried two specific choices: class-wise ranking list (i.e., each class has a ranking list) or across-class ranking list. We found that across-class ranking list works better in general, so we report the result for global ranking by default. In practice, we use top $\beta\%$ unlabeled nodes to generate the final reliable set. We do a hyperparameter search of $\beta$ from $\{60, 65, 70, 75, 80, 85, 90, 95\}$ and report the best result.
- **SW.** It is a weighted loss function version. The core idea is to assign lower sample weights to unreliable nodes while higher scores to reliable nodes. To achieve this goal, we adopt another MLPs network to learn sample weight. Specifically, the weight MLPs network takes the hidden representation learned by MLP student as input and outputs its weight score. The weight network and MLP student will be trained end-to-end by minimizing a weighted loss function.

## B.3  EVALUATION SETTINGS

Given a graph $\mathcal{G}$, the feature matrix $\mathbf{X}$, and label matrix $\mathbf{Y}$, we consider two evaluation settings in experiments: **transductive** and **inductive**. The transudtive setting is the same to standard splitting strategy, where the test data is also used during model training. In contrast, for inductive setting, we hold out a portion of test data for evaluation only. Specifically, we first pick up some inductive nodes $\mathcal{V}_{ind}^U \in \mathcal{V}^U$ from the unlabeled node set, and then partition $\mathcal{V}^U$ into two disjoint data set: inductive node set $\mathcal{V}_{ind}^U$ and observed node set $\mathcal{V}_{obs}^U$. Next, we manually remove all edges connecting to those inductive nodes and obtain two disjoint graphs $\mathcal{G} = \mathcal{G}_{obs} \cup \mathcal{G}_{ind}$. Similarly, node features and labels are partitioned into three disjoint sets $\mathbf{X} = \mathbf{X}_{obs}^U \cup \mathbf{X}_{ind}^U \cup \mathbf{X}^L$ and $\mathbf{Y} = \mathbf{Y}_{obs}^U \cup \mathbf{Y}_{ind}^U \cup \mathbf{Y}^L$. Based on these notations, the two evaluation scenarios are setup as follows:

- **Transductive:** the model is trained on $\mathcal{G}$, $\mathbf{X}$, and $\mathbf{Y}^L$, and tested on $\mathbf{X}^U$, and $\mathbf{Y}^U$, with KD based on $\mathbf{z}_v$ for $v \in \mathcal{V}$.
- **Inductive:** the model is trained on $\mathcal{G}_{obs}$, $\mathbf{X}_{obs}^U$, $\mathbf{X}^L$, and $\mathbf{Y}^L$, and tested on $\mathbf{X}_{ind}^U$ and $\mathbf{Y}_{ind}^L$, with KD based on $\mathbf{z}_v$ for $v \in \mathcal{V}_{obs}^U \cup \mathcal{V}^L$.

## B.4  IMPLEMENTATION DETAILS

To make our results reproducible, we build our model based on Pytorch and adopt the benchmark tool - PyG (PyTorch Geometric) library (Fey & Lenssen, 2019) to implement various GNNs architectures.

Table 5: Inference time (ms) on 10 randomly chosen nodes of two large-scale OGB datasets under the inductive setting.

| Dataset | SAGE | QSAGE | PSAGE | Neighbor Sample | Ours |
|---|---|---|---|---|---|
| Arxiv | 489.49 | 433.90 (1.13$\times$) | 465.43 (1.05$\times$) | 91.03 (5.37$\times$) | **3.34 (146.55$\times$)** |
| Products | 2071.30 | 1946.49 (1.06$\times$) | 2001.46 (1.04$\times$) | 107.71 (19.23$\times$) | **7.56 (273.98$\times$)** |

Following common practice in (Zhu et al., 2021b; Hu et al., 2020; 2021a), we employ a three-layer GNN encoder with dimension $d = 256$ for OGB benchmarks (ogbn-arxiv, and ogbn-products), while a two-layer GNN encoder with dimension $d = 128$ for other datasets. For MLP students, following (Zhang et al., 2022b), we set the number of layers and the hidden dimension of each layer to be the same as the teacher GNN.

For our RKD-MLP model, we do a hyperparameter search of positive augmentation ratio $\alpha_{pos}$ from $\{0.1, 0.2, \cdots, 1.0\}$, negative augmentation ratio $\alpha_{neg}$ from $\{0.1, 0.2, \cdots, 1.0\}$, learning rate from [0.01, 0.005, 0.001], weight decay from [0, 0.001, 0.002, 0.005, 0.01], and dropout from [0, 0.1, 0.2, 0.3, 0.4, 0.5, 0.6]. To accelerate the tunning process, we first fix the last three hyperparameters according to GLNN, and do grid search on $\alpha_{pos}$ and $\alpha_{neg}$. After that, we fix the two augmentation ratios, and search the learning ratio, weight decay, and dropout.

## C  EFFICIENCY ANALYSIS

To investigate the efficiency of our proposed model, we conduct experiments on two large-scale OGB datasets Arxiv and Products. Following (Zhang et al., 2022b), we include three types of baselines, i.e., vanilla GNNs, common acceleration techniques, and sampling approaches. The details are shown in the following.

**Vanilla GNNs.** We use GraphSAGE (Hamilton et al., 2017a) with GCN aggregation as a representative of conventional GNN models.

**Acceleration techniques.** Pruning and quantization are two common techniques of GNN inference acceleration. They speed up GNNs by reducing model parameters and Multiplication-and-ACcumulation (MACs) operations. In this experiment, we utilize QSAGE (quantized SAGE from FP32 to INT8) and PSAGE (SAGE with 50% weights pruned) to represent this category.

**Sampling.** Neighbor sampling is a classic approach to reducing neighbor-fetching latency for GNNs. Here, we utilize inference neighbor sampling with fan-out 15, named Neighbor Sample.

Table 5 reports the inference time of all methods under the inductive setting. From the table, we see that while other inference acceleration mechanisms could speed up SAGE, our proposed model runs significantly faster than them, let alone our better performance.

## D  MAIN RESULTS

### D.1  HOW ROBUST IS RKD-MLP W.R.T. FEATURE OR TOPOLOGY NOISES?

In this section, we conduct experiments to investigate the robustness of our RKD-MLP on two noisy situations: noisy node features and incomplete graph topology. For noisy node features, we replace $\mathbf{X}$ with $\hat{\mathbf{X}} = \mathbf{X} \odot \mathbf{M}$, where $\mathbf{M} \in \mathbb{R}^{N \times D}$ is an indicator matrix with each element is generated by a Bernoulli distribution of parameter $1 - \alpha$. For incomplete graph topology, we randomly mask each edge in $\mathcal{G}$ with probability $\alpha$, and replace the $\mathcal{G}$ with the resultant incomplete graph. Figure 5 and Figure 4 report the results of RKD-MLP on noisy features and structures, respectively, with $\alpha$ varies from 0.1 to 0.9 with step-size 0.1.

## E  MOTIVATION EXPERIMENTS

In this section, we test the impacts of noisy nodes (incorrectly predicted nodes by the teacher model) on MLP student training. Specifically, given a pre-trained GNN teacher, we first query it to give

Table 6: Node classification accuracy on commonly used graph datasets in transductive learning. *Improv.* indicates our proposal outperforms GLNN baseline. "-" indicates the result for the teacher.

| Teacher | Student | Cora | CiteSeer | PubMed | WikiCS | Compute | Photo | CS |
|---|---|---|---|---|---|---|---|---|
| GCN | - | $82.09 \pm 0.28$ | $69.62 \pm 0.28$ | $78.38 \pm 0.27$ | $67.29 \pm 0.64$ | $82.93 \pm 0.67$ | $91.09 \pm 0.49$ | $90.31 \pm 0.25$ |
| | SW | $52.24 \pm 3.40$ | $61.35 \pm 2.00$ | $75.91 \pm 1.53$ | $55.89 \pm 0.70$ | $54.01 \pm 2.04$ | $56.56 \pm 5.07$ | $35.93 \pm 1.28$ |
| | Entropy | $81.45 \pm 0.27$ | $69.95 \pm 0.49$ | $79.26 \pm 0.24$ | $70.10 \pm 0.53$ | $82.46 \pm 0.79$ | $92.78 \pm 0.45$ | $91.23 \pm 0.38$ |
| | Cluster | $74.32 \pm 0.45$ | $70.42 \pm 0.33$ | $80.90 \pm 0.14$ | $67.87 \pm 0.43$ | $81.63 \pm 0.74$ | $87.49 \pm 0.26$ | $91.66 \pm 0.13$ |
| | GLNN | $81.64 \pm 0.90$ | $69.86 \pm 0.80$ | $79.05 \pm 0.30$ | $69.43 \pm 0.83$ | $83.05 \pm 0.72$ | $92.12 \pm 0.67$ | $91.92 \pm 0.52$ |
| | RKD-MLP | $82.53 \pm 0.16$ | $71.52 \pm 0.58$ | $81.61 \pm 0.39$ | $72.13 \pm 0.65$ | $84.44 \pm 0.57$ | $93.27 \pm 0.31$ | $92.88 \pm 0.15$ |
| | *Improv.* | +1.09% | +2.37% | +3.23% | +3.88% | +1.67% | +1.24% | +1.04% |
| APPNP | - | $81.59 \pm 0.64$ | $70.44 \pm 0.21$ | $79.68 \pm 0.19$ | $67.84 \pm 1.08$ | $81.67 \pm 1.21$ | $91.92 \pm 0.95$ | $90.69 \pm 0.28$ |
| | SW | $51.02 \pm 4.11$ | $53.02 \pm 3.63$ | $77.92 \pm 0.46$ | $55.20 \pm 1.76$ | $56.06 \pm 5.88$ | $60.25 \pm 4.27$ | $35.67 \pm 0.51$ |
| | Entropy | $80.62 \pm 0.37$ | $70.37 \pm 0.61$ | $79.26 \pm 0.25$ | $68.97 \pm 1.22$ | $82.69 \pm 1.35$ | $92.16 \pm 0.40$ | $92.27 \pm 0.26$ |
| | Cluster | $75.14 \pm 0.54$ | $71.41 \pm 0.35$ | $80.59 \pm 0.25$ | $68.40 \pm 0.54$ | $81.10 \pm 0.83$ | $87.61 \pm 0.48$ | $91.64 \pm 0.16$ |
| | GLNN | $81.83 \pm 0.78$ | $70.67 \pm 0.57$ | $80.27 \pm 0.36$ | $69.87 \pm 1.03$ | $81.76 \pm 1.11$ | $91.92 \pm 1.08$ | $90.88 \pm 0.40$ |
| | RKD-MLP | $82.80 \pm 0.49$ | $71.91 \pm 0.38$ | $81.37 \pm 0.50$ | $71.44 \pm 0.56$ | $83.06 \pm 1.01$ | $93.27 \pm 0.56$ | $92.74 \pm 0.22$ |
| | *Improv.* | +1.18% | +1.75% | +1.37% | +2.24% | +1.59% | +1.46% | +2.04% |
| GAT | - | $82.64 \pm 0.63$ | $69.60 \pm 0.42$ | $77.80 \pm 0.25$ | $68.66 \pm 0.97$ | $81.90 \pm 1.51$ | $91.42 \pm 0.74$ | $89.73 \pm 0.72$ |
| | SW | $45.85 \pm 7.67$ | $47.00 \pm 13.05$ | $73.70 \pm 2.63$ | $51.82 \pm 3.56$ | $47.87 \pm 13.00$ | $62.23 \pm 7.28$ | $33.69 \pm 5.14$ |
| | Entropy | $80.67 \pm 0.69$ | $70.05 \pm 1.24$ | $79.11 \pm 0.36$ | $70.14 \pm 1.21$ | $82.26 \pm 1.73$ | $92.88 \pm 0.70$ | $89.64 \pm 2.67$ |
| | Cluster | $75.85 \pm 0.52$ | $70.39 \pm 0.52$ | $80.35 \pm 0.30$ | $67.96 \pm 0.43$ | $80.53 \pm 1.40$ | $87.46 \pm 0.59$ | $91.76 \pm 0.19$ |
| | GLNN | $82.55 \pm 0.84$ | $69.92 \pm 0.31$ | $78.81 \pm 0.45$ | $70.69 \pm 1.17$ | $82.19 \pm 1.38$ | $91.67 \pm 0.62$ | $91.00 \pm 0.91$ |
| | RKD-MLP | $84.12 \pm 0.36$ | $72.56 \pm 0.68$ | $82.19 \pm 0.27$ | $73.21 \pm 0.84$ | $83.85 \pm 0.53$ | $93.49 \pm 0.66$ | $92.77 \pm 0.20$ |
| | *Improv.* | +1.90% | +3.77% | +4.28% | +3.56% | +2.01% | +1.98% | +1.94% |
| SGC | - | $80.59 \pm 0.47$ | $69.59 \pm 0.31$ | $78.30 \pm 0.14$ | $67.78 \pm 0.59$ | $82.70 \pm 0.64$ | $91.30 \pm 0.51$ | $90.13 \pm 0.38$ |
| | SW | $46.80 \pm 4.24$ | $45.11 \pm 14.58$ | $75.84 \pm 1.05$ | $60.44 \pm 1.90$ | $54.11 \pm 6.35$ | $59.29 \pm 3.59$ | $35.98 \pm 1.00$ |
| | Entropy | $80.40 \pm 0.40$ | $70.34 \pm 0.64$ | $79.54 \pm 0.15$ | $69.79 \pm 0.52$ | $82.49 \pm 1.17$ | $92.22 \pm 0.81$ | $91.85 \pm 0.48$ |
| | Cluster | $74.60 \pm 0.99$ | $70.08 \pm 0.70$ | $80.69 \pm 0.19$ | $67.62 \pm 0.91$ | $81.60 \pm 0.75$ | $87.32 \pm 0.70$ | $91.90 \pm 0.15$ |
| | GLNN | $80.93 \pm 0.75$ | $69.87 \pm 0.32$ | $78.99 \pm 0.16$ | $69.87 \pm 0.56$ | $83.08 \pm 0.70$ | $92.54 \pm 0.37$ | $91.39 \pm 0.65$ |
| | RKD-MLP | $82.99 \pm 0.34$ | $71.28 \pm 0.92$ | $81.72 \pm 0.40$ | $72.56 \pm 0.83$ | $84.56 \pm 0.89$ | $93.53 \pm 0.46$ | $93.02 \pm 0.40$ |
| | *Improv.* | +2.54% | +2.01% | +3.45% | +3.85% | +1.78% | +1.06% | +1.78% |

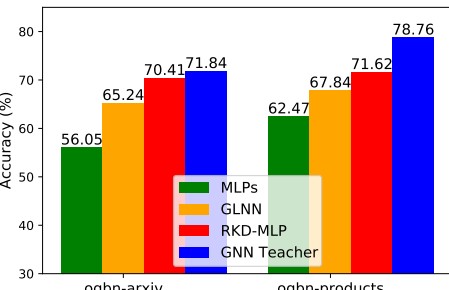

Figure 7: Accuracy results of RKD-MLP on large-scale graphs. **Left**: GCN teacher. **Right**: GraphSAINT teacher.

predictions for all nodes (including both labeled and unlabeled nodes). Then we divide nodes into two categories based on whether they are correctly predicted by the teacher model or not. We use $\mathcal{S}_{correct}$ and $\mathcal{S}_{wrong}$ to denote the correctly and incorrectly predicted node sets, respectively. Let $\omega$ be the noisy ratio, which is selected from $\{0.0, 0.05, 0.1, 0.15, 0.2, 0.25, 0.30\}$. For a given $\omega$, we randomly sample $|\mathcal{S}_{correct}| * \omega$ nodes from the $\mathcal{S}_{wrong}$ set and extend them to the correctly predicted set, resulting a noisy node set $\hat{\mathcal{S}}$. After that, we train the MLP student based on $\hat{\mathcal{S}}$ and report the averaged results of ten runs. Figure 8 summarizes the results on Cora, WikiCS and ogbn-products datasets.

The interesting observation is that the GNN teacher outperforms the MLP student even in 0% noisy scenario, which is the upper bound the MLP student could achieve under KD. This result

Table 7: Node classification accuracy on commonly used graph datasets in inductive learning. *Improv.* indicates our proposal outperforms GLNN baseline. "-" indicates the result for the teacher.

| Teacher | Student | Cora | CiteSeer | PubMed | WikiCS | Compute | Photo | CS |
|---|---|---|---|---|---|---|---|---|
| MLPs | - | $59.09 \pm 2.96$ | $59.60 \pm 1.57$ | $67.73 \pm 1.23$ | $62.55 \pm 1.85$ | $67.84 \pm 1.78$ | $79.44 \pm 1.72$ | $85.57 \pm 0.92$ |
| SAGE | - | $80.00 \pm 0.42$ | $71.79 \pm 0.22$ | $77.50 \pm 0.19$ | $65.58 \pm 1.47$ | $75.53 \pm 1.61$ | $87.13 \pm 0.34$ | $91.31 \pm 0.36$ |
| | SW | $45.81 \pm 2.68$ | $52.05 \pm 17.53$ | $78.56 \pm 0.87$ | $56.32 \pm 2.34$ | $46.74 \pm 13.93$ | $64.22 \pm 6.53$ | $34.60 \pm 2.27$ |
| | Entropy | $81.02 \pm 0.36$ | $73.20 \pm 0.58$ | $79.74 \pm 0.37$ | $67.78 \pm 0.93$ | $80.95 \pm 0.83$ | $91.35 \pm 0.49$ | $91.37 \pm 0.10$ |
| | Cluster | $72.98 \pm 0.73$ | $74.86 \pm 0.80$ | $80.20 \pm 0.41$ | $66.36 \pm 0.57$ | $81.25 \pm 1.10$ | $87.57 \pm 0.44$ | $91.24 \pm 0.11$ |
| | GLNN | $80.19 \pm 0.71$ | $72.31 \pm 0.58$ | $78.76 \pm 0.58$ | $67.29 \pm 1.31$ | $76.30 \pm 1.65$ | $87.70 \pm 0.40$ | $91.38 \pm 0.53$ |
| | RKD-MLP | $81.69 \pm 0.96$ | $75.00 \pm 0.29$ | $81.18 \pm 0.65$ | $69.62 \pm 0.43$ | $81.50 \pm 0.77$ | $91.58 \pm 1.45$ | $92.41 \pm 0.54$ |
| | *Improv.* | +1.87% | +3.72% | +3.07% | +3.46% | +6.81% | +4.42% | +1.12% |
| APPNP | - | $82.87 \pm 1.06$ | $72.50 \pm 0.71$ | $79.18 \pm 0.12$ | $67.33 \pm 0.88$ | $80.24 \pm 0.62$ | $77.30 \pm 1.94$ | $89.61 \pm 0.22$ |
| | SW | $47.89 \pm 3.89$ | $59.22 \pm 4.55$ | $78.32 \pm 0.94$ | $54.46 \pm 2.72$ | $56.82 \pm 4.46$ | $61.88 \pm 2.51$ | $35.81 \pm 2.63$ |
| | Entropy | $79.61 \pm 1.63$ | $74.23 \pm 0.64$ | $80.54 \pm 0.43$ | $68.47 \pm 0.84$ | $81.20 \pm 0.23$ | $82.88 \pm 0.84$ | $91.79 \pm 0.17$ |
| | Cluster | $73.27 \pm 0.52$ | $74.23 \pm 1.09$ | $80.36 \pm 0.42$ | $66.21 \pm 0.41$ | $79.75 \pm 0.81$ | $80.61 \pm 1.54$ | $91.25 \pm 0.10$ |
| | GLNN | $82.68 \pm 0.68$ | $72.75 \pm 1.08$ | $79.88 \pm 0.23$ | $68.78 \pm 1.00$ | $80.22 \pm 0.54$ | $78.08 \pm 1.91$ | $89.99 \pm 0.29$ |
| | RKD-MLP | $83.38 \pm 1.28$ | $74.93 \pm 0.79$ | $81.44 \pm 0.51$ | $69.83 \pm 1.01$ | $82.60 \pm 1.24$ | $85.10 \pm 0.90$ | $92.60 \pm 0.19$ |
| | *Improv.* | +0.84% | +2.99% | +1.95% | +1.52% | +2.96% | +8.99% | +2.90% |
| GAT | - | $81.07 \pm 0.79$ | $72.72 \pm 0.60$ | $78.00 \pm 0.28$ | $67.77 \pm 1.19$ | $80.20 \pm 1.38$ | $90.38 \pm 1.02$ | $89.24 \pm 0.86$ |
| | SW | $43.68 \pm 3.29$ | $50.13 \pm 9.22$ | $76.98 \pm 2.02$ | $54.50 \pm 2.58$ | $57.40 \pm 3.04$ | $59.34 \pm 4.60$ | $34.44 \pm 1.76$ |
| | Entropy | $81.89 \pm 1.36$ | $74.09 \pm 0.60$ | $78.56 \pm 1.13$ | $69.99 \pm 1.51$ | $81.13 \pm 2.40$ | $91.83 \pm 1.24$ | $90.64 \pm 0.65$ |
| | Cluster | $73.85 \pm 0.93$ | $74.53 \pm 0.57$ | $80.36 \pm 0.15$ | $66.54 \pm 0.50$ | $80.58 \pm 0.16$ | $88.33 \pm 0.57$ | $90.15 \pm 2.61$ |
| | GLNN | $81.89 \pm 0.39$ | $72.94 \pm 0.73$ | $79.68 \pm 0.62$ | $70.10 \pm 1.41$ | $80.69 \pm 0.26$ | $91.31 \pm 1.08$ | $90.97 \pm 0.97$ |
| | RKD-MLP | $82.61 \pm 1.30$ | $74.86 \pm 0.58$ | $82.28 \pm 0.40$ | $71.54 \pm 0.33$ | $81.97 \pm 0.39$ | $92.41 \pm 0.90$ | $92.86 \pm 0.26$ |
| | *Improv.* | +0.88% | +2.63% | +3.26% | +2.05% | +1.58% | +1.20% | +2.08% |
| SGC | - | $66.15 \pm 0.58$ | $66.47 \pm 0.74$ | $75.60 \pm 0.59$ | $60.62 \pm 0.26$ | $72.70 \pm 1.56$ | $81.10 \pm 0.75$ | $87.03 \pm 0.38$ |
| | SW | $45.86 \pm 8.28$ | $46.77 \pm 3.72$ | $76.44 \pm 1.98$ | $57.01 \pm 1.09$ | $48.21 \pm 8.94$ | $63.64 \pm 1.47$ | $34.14 \pm 2.34$ |
| | Entropy | $68.23 \pm 0.82$ | $69.83 \pm 0.99$ | $78.32 \pm 0.64$ | $65.05 \pm 1.18$ | $78.11 \pm 0.65$ | $87.52 \pm 0.64$ | $89.98 \pm 0.75$ |
| | Cluster | $70.07 \pm 0.66$ | $71.39 \pm 0.63$ | $80.42 \pm 0.19$ | $65.42 \pm 0.75$ | $80.15 \pm 0.70$ | $84.56 \pm 0.35$ | $90.35 \pm 0.07$ |
| | GLNN | $66.54 \pm 0.71$ | $66.65 \pm 0.36$ | $76.70 \pm 0.41$ | $63.14 \pm 1.12$ | $75.48 \pm 1.27$ | $84.83 \pm 2.60$ | $88.09 \pm 0.66$ |
| | RKD-MLP | $71.08 \pm 1.00$ | $71.45 \pm 1.49$ | $81.70 \pm 0.26$ | $66.04 \pm 0.88$ | $81.23 \pm 0.93$ | $89.03 \pm 0.68$ | $92.75 \pm 0.69$ |
| | *Improv.* | +6.82% | +7.20% | +6.52% | +4.59% | +7.62% | +4.95% | +5.29% |

Table 8: Hyperparameters of different GNN teachers on small datasets: Cora, CiteSeer, PubMed, WikiCS, Amazon-Computers, Amazon-Photo, and Coauthor-CS.

| | SAGE | GCN | GAT | APPNP | SGC |
|---|---|---|---|---|---|
| # layers | 2 | 2 | 2 | 2 | 2 |
| hidden dim | 128 | 128 | 128 | 128 | 128 |
| learning rate | 0.01 | 0.01 | 0.01 | 0.01 | 0.1 |
| weight decay | 0.0005 | 0.001 | 0.01 | 0.01 | 0.001 |
| dropout | 0.0 | 0.8 | 0.6 | 0.5 | 0.0 |
| attention heads | - | - | 8 | - | - |
| power iterations | - | - | - | 10 | - |

calls for more advanced KD techniques (in addition to distill soft labels) to further bridge the performance gap between MLP student and GNN model on extremely large datasts. However, the promising observation is that, by removing those noisy nodes, the noisy-aware MLP student could consistently achieve better results than the strong baseline – GLNN. It further confirms our motivation in considering noise-aware MLP student training in this paper.

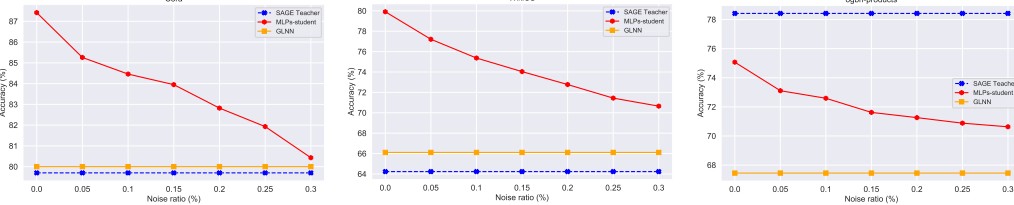

Figure 8: The impacts of incorrectly predicted nodes on the MLP student under different noise ratios.

Table 9: Hyperparameters of GraphSAGE and GCN on ogbn-arxiv dataset, while ClusterGCN and GraphSAINT on ogbn-products dataset according to (Hu et al., 2021a)

|  | SAGE | GCN | ClusterGCN | GraphSAINT |
|---|---|---|---|---|
| # layers | 3 | 3 | 3 | 3 |
| hidden dim | 256 | 256 | 256 | 256 |
| learning rate | 0.01 | 0.01 | 0.001 | 0.01 |
| dropout | 0.5 | 0.5 | 0.5 | 0.5 |
| partition number | - | - | 15,000 | - |
| batch size | - | - | 32 | 20,000 |

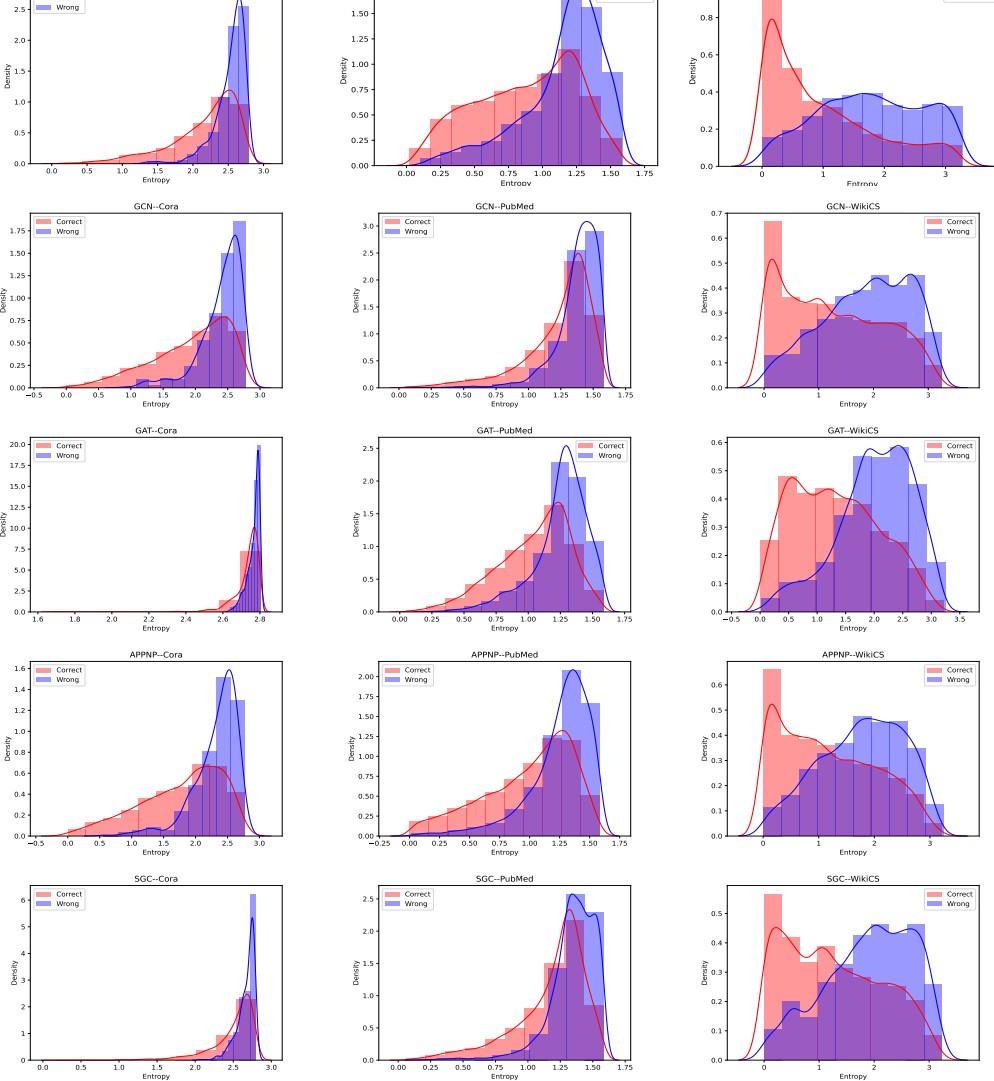

Figure 9: The entropy distributions of wrongly and correctly predicted groups generated by 5 GNN backbones on Cora, PubMed, and WikiCS datasets. Each row presents one GNN model.

## F  PERFORMANCE ON GNN STUDENT MODELS

In this section, we test the applicability of our proposal over GNN student models. Specifically, follow previous experimental protocols, we adopt GNN model as student, which has the same architecture with the GNN teacher accordingly. Table 10 reports the results under transductive setting. From

Table 10: Node classification accuracy on commonly used graph datasets in transductive learning. The student model is GNN architecture.

| Teacher | Student | Cora | CiteSeer | PubMed | WikiCS | Compute | Photo | CS |
|---|---|---|---|---|---|---|---|---|
| SAGE | - | $79.70 \pm 0.52$ | $68.59 \pm 0.27$ | $76.55 \pm 0.29$ | $65.59 \pm 0.88$ | $82.97 \pm 2.16$ | $90.90 \pm 0.84$ | $90.56 \pm 0.38$ |
| | SAGE | $83.17 \pm 0.77$ | $71.86 \pm 0.16$ | $79.54 \pm 0.62$ | $70.35 \pm 1.03$ | $84.34 \pm 0.55$ | $93.55 \pm 0.14$ | $92.11 \pm 0.01$ |
| GCN | - | $82.09 \pm 0.28$ | $69.62 \pm 0.28$ | $78.38 \pm 0.27$ | $67.29 \pm 0.64$ | $82.93 \pm 0.67$ | $91.09 \pm 0.49$ | $90.31 \pm 0.25$ |
| | GCN | $83.94 \pm 0.67$ | $72.03 \pm 0.00$ | $80.30 \pm 0.00$ | $70.20 \pm 0.56$ | $83.70 \pm 0.70$ | $91.98 \pm 0.21$ | $92.17 \pm 0.14$ |
| APPNP | - | $81.59 \pm 0.64$ | $70.44 \pm 0.21$ | $79.68 \pm 0.19$ | $67.84 \pm 1.08$ | $81.67 \pm 1.21$ | $91.92 \pm 0.95$ | $90.69 \pm 0.28$ |
| | APPNP | $83.77 \pm 0.70$ | $72.70 \pm 0.18$ | $81.43 \pm 0.49$ | $72.13 \pm 0.52$ | $83.58 \pm 0.12$ | $92.65 \pm 0.44$ | $93.23 \pm 0.07$ |
| GAT | - | $82.64 \pm 0.63$ | $69.60 \pm 0.42$ | $77.80 \pm 0.25$ | $68.66 \pm 0.97$ | $81.90 \pm 1.51$ | $91.42 \pm 0.74$ | $89.73 \pm 0.72$ |
| | GAT | $84.40 \pm 0.07$ | $71.85 \pm 0.07$ | $79.84 \pm 0.46$ | $71.98 \pm 0.01$ | $83.13 \pm 0.22$ | $93.02 \pm 0.31$ | $90.85 \pm 0.84$ |
| SGC | - | $80.59 \pm 0.47$ | $69.59 \pm 0.31$ | $78.30 \pm 0.14$ | $67.78 \pm 0.59$ | $82.70 \pm 0.64$ | $91.30 \pm 0.51$ | $90.13 \pm 0.38$ |
| | SGC | $83.82 \pm 0.07$ | $71.57 \pm 0.46$ | $80.44 \pm 0.07$ | $70.09 \pm 0.17$ | $84.05 \pm 0.43$ | $92.47 \pm 0.45$ | $92.11 \pm 0.01$ |

the results, we can observer that the GNN students could perform generally better than their teacher models acorss 7 datasets. These results further verify the effectiveness of our meta-policy in achieving reliable knowledge distillation.

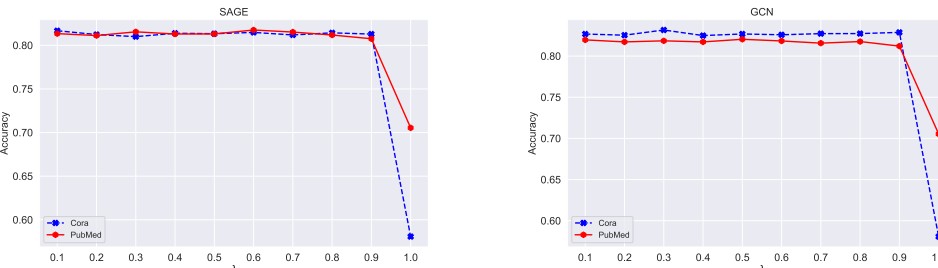

Figure 10: The impacts of trade-off parameter $\lambda$ on RKD-MLP.

## G  THE IMPACTS OF $\lambda$

In this section, we analyze the impacts of $\lambda$ on our RKD-MLP model. Specifically, we vary $\lambda$ from 0.1 to 1.0 with step size 0.1 and report the results on Figure 10.

From the figures, we can observe that our RKD-MLP model performs generally stable when $0.1 \leq \lambda \leq 0.9$. When $\lambda = 1.0$, it reduces to the vanilla MLP baseline, so the performance drops significantly.

