# OpenReview forum: "Double Wins: Boosting Accuracy and Efficiency of Graph Neural Networks by Reliable Knowledge Distillation"
_ICLR.cc/2023/Conference — Submitted to ICLR 2023_

### Official Review · Reviewer_GWUx · 2022-10-21

**Confidence:** 4
**Correctness:** 3
**Technical Novelty And Significance:** 3
**Empirical Novelty And Significance:** 3
**Recommendation:** 6

**Clarity, Quality, Novelty And Reproducibility:**

The proposed method is well-explained, and all the implementations detail, including source code, are included.

**Details Of Ethics Concerns:**

I haven't found any ethical issues.

**Strength And Weaknesses:**

*Strength*

The paper is well-written, and the method is well-explained.
The evaluation of the proposed is very impressive, including various datasets under both transductive and inductive settings
The authors performed an extension ablation study of the proposed dataset, showing how different model parameter choices impact performance.

*Weakness*
My major concern is why the authors haven't evaluated their method with SOTA architectures GNNs as teachers.
Please see the paperwithcode SOTA benchamrks.





**Summary Of The Paper:**

This paper proposed Knowledge Distillation (KD) method for Graph Neural Networks.
The proposed method is motivated by a recent study that shows that  GNNs can be compressed to inference-friendly multi-layer perceptrons (MLPs) by training MLPs using the soft labels of labeled and unlabeled nodes from the teacher.
However, leveraging the soft labels of all unlabeled nodes may be suboptimal since the teacher model would inevitably make wrong predictions.
This paper's proposed method, called: Reliable Knowledge Distillation for MLP optimization, is the first noise-aware knowledge distillation framework for GNNs. Its core idea is to use a meta-policy to filter out those unreliable soft labels. For training a meta-policy, the authors propose a reward-driven objective based on a meta-set and adopt a policy gradient to optimize the expected reward.
Afterward, the authors use the meta-policy to the unlabeled nodes and select the most reliable soft labels for distillation.
The extensive experiments across various GNN models,  7 small graphs, and 2 large-scale datasets demonstrate the superiority of the proposed method to the prior art.


**Summary Of The Review:**

This paper proposed Knowledge Distillation (KD) method for Graph Neural Networks.
The proposed method is motivated by a recent study that shows that  GNNs can be compressed to inference-friendly multi-layer perceptrons (MLPs) by training MLPs using the soft labels of labeled and unlabeled nodes from the teacher.
The method is well-explained, and evaluations are solid. My main concern is that there are still hand-crafted GNN-based architectures that outperform the proposed method's performances. See the SOTA result on paperwithcodes. I would like the authors to elaborate on this regard. Specifically, the authors conclude from the results of table 10 that they manage to achieve better performance of the student rather than the performance of the teacher. Are the same phenomena would preserve also using SOTA backbones as a teacher?

In overall, I am positive about the novelty and the impact of the proposed method, looking forward receiving responce on the raised issue.

---

> ### Author Response · Authors · 2022-11-18
> **Response to Comments from Reviewer GWUx**
>
> We greatly thank the reviewer GWUx for his/her helpful and insightful comments. We provide our responses to the comments as follows.
>
> **Q1.My major concern is why the authors haven't evaluated their method with SOTA architectures GNNs as teachers. Please see the paper with code SOTA benchmarks. I would like the authors to elaborate on this regard. Specifically, the authors conclude from the results of table 10 that they manage to achieve better performance of the student rather than the performance of the teacher. Are the same phenomena preserved also using SOTA backbones as a teacher?**
>
> Thank you for raising the concern. The strong student performance can be attributed to the policy network since it can filter out unreliable nodes. In paper with code, the SOTA algorithms are different in different benchmark datasets. Some of the results in paper with code even used different data splits, so their results are not comparable at all. To rigorously evaluate our method, we used the standard GNNs as the backbones and standard data splits.We have demonstrated consistent improvements on GCN, SAGE, GAT, APPNP, and SGC.
>
> Following your suggestions, we consider CPF [1] as an additional SOTA baseline since it has exactly the same data splits as ours. Specifically, we select the CPF-GCN version since it shows the best results across different datasets on average. Then we apply GLNN and our RKD-MLP with CPF as teachers. The results are listed below.
> |         | Cora   | CiteSeer | PubMed | A-Computers | A-Photo |
> |---------|--------|----------|--------|-------------|---------|
> | CPF     | 0.8467 | 0.7582   | 0.8084 | 0.8373      | 0.9245  |
> | GLNN (CPF Teacher)    | 0.8504 | 0.7638   | 0.8105 | 0.8413      | 0.9266  |
> | RKD-MLP (CPF Teacher) | 0.8645 | 0.7753   | 0.8234 | 0.8567      | 0.9385  |
>
> While CPF performs better than three standard GNNs in our paper, the performance of RKD-MLP also improves using CPF as a teacher. This comparison suggests the effectiveness and generalizability of our distillation strategy.
>
> [1]. Yang, Cheng, et al. Extract the knowledge of graph neural networks and go beyond it: An effective knowledge distillation framework, www, 2021.

---

### Official Review · Reviewer_aYE5 · 2022-10-24

**Confidence:** 4
**Correctness:** 3
**Technical Novelty And Significance:** 2
**Empirical Novelty And Significance:** 2
**Recommendation:** 3

**Clarity, Quality, Novelty And Reproducibility:**

The proposed method is quite straightforward, and it is incremental work following the GLNN. I think the novelty is limited.

Table 1 and Table 2 is not very friendly for readers. The best performance is not bold, and the second best is not indicated. Which may mislead readers in terms of the improvement. Moreover, the caption is saying that the improvement is calculated by comparing with GLNN only, which does not very make sense to me. I understand that as a follow-up work, compare with GLNN is necessary. But when we say the improvement, we normally will report the improvement achieved compared with the 2nd best performance.

As it is clear that on Table 1, the cluster approach is very close to the proposed RKD-MLP (i.e., 80.66 vs 80.97, 0.38% improvement).

I do not agree to place the main experiments into the Appendix especially when you place the keyword on your title.

**Details Of Ethics Concerns:**

N.A.

**Strength And Weaknesses:**

Strengths:
1. It is an interesting approach that aims to achieve better accuracy and efficiency for GNN.
2. The idea is intuitionistic and easy to follow and understand.

Weaknesses:
1. As outlined in the title, the proposed method aims to boost accuracy and efficiency. However, the crucial experiments on the efficiency are placed in the appendix which is not reader-friendly. If I only go through the main context of this paper, I will have no idea how the proposed method can boost efficiency. There is no clue that why the proposed method can achieve good efficiency. It takes too many spaces to report the accuracy results in your main text, and some of them should be swapped with the efficiency results in your Appendix.
2. As the authors mentioned in the paper, the meta-set is the validation set. How the dataset is split? How does the size of the "meta-set" (i.e., validation set) affect your performance in terms of accuracy and efficiency? I think this question is worth investigating and quite important. If the proposed model can achieve a better performance than existing methods by using a small "meta-set", it can provide some evidence of the efficiency.


**Summary Of The Paper:**

This paper aims to design a model which can achieve better accuracy and efficiency. The proposed method RKD-MLP utilizes knowledge distillation techniques and the idea similar to noise-label learning to achieve the mentioned goals. RKD-MLP aims to learn the 'reliable' soft label (which is called meta-policy in this paper) to guide the student network in learning a better MLP network.

Different from existing works, this paper is using the joint training approach to train the network under the reinforcement learning framework i.e., policy gradient with advantage.

**Summary Of The Review:**

Overall, considering the above points, especially the novelty and experiments, I think this paper is not good enough for acceptance.

---

> ### Author Response · Authors · 2022-11-18
> **Response to Comments from Reviewer aYE5  (Part 1)**
>
> We greatly thank the reviewer aYE5 for his/her helpful and insightful comments. We provide our responses to the comments as follows.
>
> **Q1.It takes too many spaces to report the accuracy results in your main text, and some of them should be swapped with the efficiency results in your Appendix.**
>
> Thank you for pointing this out. In the revised version, we have moved the efficiency analysis to the main body and reported the results in Table 3 as follows.
> | Dataset  | SAGE    | QSAGE           | PSAGE           | Neighbor Sample | Ours           |
> |----------|---------|-----------------|-----------------|-----------------|----------------|
> | Arxiv    | 489.49  | 433.90 (1.13x)  | 465.43 (1.05x)  | 91.03 (5.37x)   | 3.34 (146.55x) |
> | Products | 2071.30 | 1946.49 (1.06x) | 2001.46 (1.04x) | 107.71 (19.23x) | 7.56 (273.98x) |
>
> We formally analyzed the efficiency of our model in RQ6 of Section 4.2. Specifically, our method RKD-MLP is significantly faster than vanilla GNNs and other acceleration baselines based on pruning, quantization, and sampling. This is because our method RKD-MLP reduces to simple MLPs after distillation, so we can directly feed the features of the query node to the trained MLPs for inference. However, all baseline methods still require sampling and fetching the neighboring nodes of the query node to perform graph convolution, which is time-consuming and costly.
>
> **Q2. The meta-set is the validation set. How is the dataset split?**
>
> In our experiments, we use the splitting strategy in GLNN to split the data. Specifically, for the seven small datasets (Cora, CiteSeer, PubMed, WikiCS, Computer, Photo, CS), we randomly split 20 nodes from each class as labeled nodes for training, 30 nodes for validation, and all other nodes for testing. For OGB datasets, we follow the OGB official splits for Arxiv and Products, which results in 54%, 18%, and 28% nodes for training, validation, and testing on ogbn-arxiv, and 8%, 2%, and 90% nodes for training, validation, and testing on ogbn-products, respectively.
>
>
> **Q3. How does the size of the "meta-set" (i.e., validation set) affect your performance in terms of accuracy and efficiency? I think this question is worth investigating and quite important. If the proposed model can achieve a better performance than existing methods by using a small "meta-set", it can provide some evidence of the efficiency.**
>
> Following your suggestions, we conduct experiments on the orgn-arxiv dataset to test the impact of meta-set size on RKD-MLP based on SAGE teacher. Specifically, the standard splitting ratios for training, validation, and testing are 54%, 18%, and 28%, respectively. We randomly sample some nodes in the validation set to construct $\omega %$ of the validation set in total. The results are listed below.
> | $\omega%$ | 1%    | 5%    | 10%   | 15%   | 18%   |
> |----------|-------|-------|-------|-------|-------|
> | RKD-MLP  | 70.35 | 70.37 | 70.43 | 70.45 | 70.49 |
>
> From the table, we can see that the size of the validation set in ogbn-arxiv has negligible results on RKD-MLP. The possible result is that even 1% nodes in ogbn-arxiv still contain enough positive and negative samples for the policy training.
>
> To verify this point, we directly use the training set as a meta-set for policy training. We found that the test accuracy is 70.64. The primary result is that the training accuracy of SAGE on ogbn-arxiv is around 78%. That is, 22% of the nodes will be wrongly classified by the SAGE teacher, so the meta-set still has positive and negative samples for training. These observations verify the promise of our meta-policy because when the GNN model does not over-fit the training set, we may directly use the training set to guide the learning of meta-policy, reducing the reliance on additional validation sets.

---

> > ### Author Response · Authors · 2022-11-18
> > **Response to Comments from Reviewer aYE5 (Part 2)**
> >
> > Sorry for splitting the responses into two parts because of the limits on the character number. Please read the left responses as follows:
> >
> > **Q4.The proposed method is quite straightforward, and it is incremental work following the GLNN. I think the novelty is limited.**
> >
> > We thank the reviewer for the feedback. We summarize the novelties of our work compared with the existing studies below.
> >
> > **Problem novelty.** How to perform effective knowledge distillation under semi-supervised scenarios is an important yet under-explored problem in both graph and vision domains. Prior distillation efforts (e.g., GLNN) mainly focus on using the soft labels of few labeled and abundant unlabeled nodes to guide the training of the MLP student, which will lead to accuracy drop because of the limited predictive ability of GNNs under semi-supervised learning. In contrast, we explored an orthogonal direction by explicitly filtering out wrongly classified unlabeled nodes and then training the MLP student using the remaining nodes, which contains less noise.
> >
> > **Technical novelty.** Unlike GLNN which directly applies traditional knowledge distillation loss function to semi-supervised scenarios, we develop a tailored distillation algorithm to tackle the noisy issue incurred by wrongly predicted unlabeled nodes from the teacher. From the technical perspective, RKD-MLP makes the following major contributions w.r.t. GLNN.
> >
> > 1. We formulate the node filtering task as a reinforcement learning problem, where an agent takes actions to decide whether each node should be kept. The agent is trained to maximize the expected reward based on the groundtruths in the meta-set to explicitly identify the reliable nodes. This formulation is a novel contribution.
> >
> > 2. The meta-policy can implicitly decide how many nodes should be filtered. The uncertainty-based methods (e.g., entropy) score each node and use a threshold to decide the percentage of the nodes to be filtered out, which introduces an additional hyperparameter. In contrast, our method encodes the filtering decisions as actions so that it does not require a threshold. This end-to-end optimization strategy is innovative.
> >
> > **Q5.Table 1 and Table 2 are not very friendly for readers. The best performance is not bold, and the second best is not indicated. Which may mislead readers in terms of the improvement.**
> >
> > Thank you for pointing this out. In the revised manuscript, the best and second-best results are highlighted in Bold font and underlined, respectively.
> >
> > **Q6.I do not agree to place the main experiments into the Appendix especially when you place the keyword on your title.**
> >
> > Thank you for the suggestion. In the updated version, we have moved the efficiency experiments to Table 3 of the main body and analyzed the results in the RQ6 of Section 4.2.

---

### Official Review · Reviewer_asz6 · 2022-10-24

**Confidence:** 4
**Correctness:** 3
**Technical Novelty And Significance:** 3
**Empirical Novelty And Significance:** 3
**Recommendation:** 6

**Clarity, Quality, Novelty And Reproducibility:**

The paper has good clarity, quality, novelty, and reproducibility.

Minor:
1. More explanation and discussion about Figure 4 would be desirable.
2. It would be interesting to discuss or compare the inference speed between pruning, quantification, and your method.
3. There are some typos, for example in the related work section “However,s it only”.


**Strength And Weaknesses:**

Strength:
1. They study the impact of unlabeled nodes in GNNs distillation and demonstrate the importance of reliable pseudo labeling.
2. They provide a simple RL-based method to filter out unlabeled nodes with unreliable predicted soft labels.
3. Experiments show that the new method can improve the accuracy of different datasets.

Weakness:
1. Including more analysis on whether the meta-policy can filter out unreliable nodes would be better. Figure 6 shows some results. But the GNN prediction is used as the ground truth, which may be incorrect. One possible way is to test the policy on some labeled nodes in the testing set.
2. I would like to know how many unlabeled nodes are removed by the meta-policy. If too many nodes are removed, we may lose too much information from the data. The meta-policy uses 0.5 as the threshold, have you tried other thresholds?
3. This framework is doing something similar to out-of-distribution detection. It would be great to provide some discussion on connections or performance comparisons between the two.


**Summary Of The Paper:**

This paper proposes a new semi-supervised KD framework (RKD-MLP) to filter out unlabeled nodes whose soft labels are likely to be incorrectly predicted by teachers. In this framework, they use a simple reinforcement learning framework to learn whether nodes have reliable soft labels, thus improving performance and maintaining inference efficiency. The experiments show the benefits of RKD-MLP.

**Summary Of The Review:**

Overall, this paper has a clear motivation and proposes an effective method. But in the experiment, most of the results are about the performance of GNNs students. It would be nice to have more results about the meta-policy.

---

> ### Author Response · Authors · 2022-11-18
> **Response to Comments from Reviewer asz6 (Part 1)**
>
> We greatly thank the reviewer asz6 for his/her helpful and insightful comments. We provide our responses to the comments as follows.
>
> **Q1. Including more analysis on whether the meta-policy can filter out unreliable nodes would be better. Figure 6 shows some results. But the GNN prediction is used as the ground truth, which may be incorrect. One possible way is to test the policy on some labeled nodes in the testing set.**
>
> Thank you for pointing this out. Following your suggestions, we have updated Figure 6 in the revised manuscript. Instead of reporting the confusion matrix, we visualize the meta-policy’s decisions w.r.t. the GNN teacher on the unlabeled nodes (a.k.a. testing set in the transductive setting) from two groups, where group 0 and 1 indicate the GNN teacher makes the right and wrong predictions, respectively. In light of this, we can clearly understand how many nodes being wrongly (or correctly) classified by the GNN teacher are filtered out (or preserved) by the proposed meta-policy. For example, from the ogbn-arxiv result in Figure 6, we observe that RKD-MLP detects 93.34% (66.89/71.66) unlabeled nodes being correctly predicted by the teacher while filtering out 67.96% (19.26/28.34) nodes being wrongly predicted on the ogbn-arxiv dataset. This analysis is updated in RQ4 of Section 4.2.
>
> **Q2.I would like to know how many unlabeled nodes are removed by the meta-policy. If too many nodes are removed, we may lose too much information from the data. The meta-policy uses 0.5 as the threshold, have you tried other thresholds?**
>
> Thank you for the question. We updated Figure 6 in the revised manuscript to make the contribution of our meta-policy clear. In Figure 6, we visualize the meta-policy’s decisions w.r.t. the GNN teacher on the unlabeled nodes (a.k.a. testing set in the transductive setting) from two groups, where groups 0 and 1 indicate the GNN teacher makes the right and wrong predictions, respectively. In general, we can observe that RKD-MLP can effectively detect nodes correctly predicted by the GNN teacher as much as possible while filtering out nodes being wrongly classified to some extent. Specifically, from the ogbn-arxiv result in Figure 6, we observe that RKD-MLP detects 93.34% (66.89/71.66) unlabeled nodes being correctly predicted by the teacher while filtering out 67.96% (19.26/28.34) nodes being wrongly predicted on the ogbn-arxiv dataset.
>
> For the 0.5 threshold, we believe there is a misunderstanding. The filtering decision is made by the policy. We use 0.5 because there are two actions for the policy, i.e., 1 for keeping and 0 for removing. So a threshold of 0.5 essentially means the probability of 1 is larger than 0. Since in reinforcement learning, we often choose argmax of the probabilities so we use 0.5. To avoid confusion, we have replaced 0.5 with $\pi_\theta(a=0|\mathbf{x}_v)$.
>
> **Q3.This framework is doing something similar to out-of-distribution detection. It would be great to provide some discussion on connections or performance comparisons between the two.**
>
> Thank you for pointing out that the method is similar to out-of-distribution detection. While out-of-distribution identifies the unreliable instances in an unsupervised way, our method learns a policy with reinforcement learning to identify the unreliable instances using the feedback from a meta-set, which can directly maximize the number of correct predictions. To address your concern, we have implemented an auto-encoder for out-of-distribution detection. Specifically, we feed node features to an MLP encoder to map nodes into hidden space, and then reconstruct the input features via another decoder network. After the model is trained, we compute the reconstruction losses of nodes and use them to determine noisy nodes according to their ranks. For a fair comparison, we search the encoder and decoder layers from {1,2,3,4,5}, hidden dimensions from {64, 128, 256, 512}, and top ranking ratio from {1%, 5%, 10%, 15%, 20%, 25%} which are used to determine noisy nodes. Due to time limitations, we tested the results based on the SAGE teacher as follows.
> |             | Cora  | CiteSeer | PubMed | WikiCS | Compute | Photo | CS    | ogbn-arxiv |
> |-------------|-------|----------|--------|--------|---------|-------|-------|------------|
> | Autoencoder | 61.25 | 63.74    | 74.72  | 62.55  | 64.48   | 83.36 | 88.87 | 58.67      |
>
> From the table above, we observe that the performance of standard autoencoder-based methods is not good because it is a purely unsupervised approach and cannot utilize the graph topology. In addition, we have added additional related work in Appendix H to discuss the relationships with the out-of-distribution methods.

---

> > ### Author Response · Authors · 2022-11-18
> > **Response to Comments from Reviewer asz6 (Part 2)**
> >
> > Sorry for splitting the responses into two parts because of the limits on the character number. Please read the left responses as follows:
> >
> > **Q4.More explanation and discussion about Figure 4 would be desirable.**
> >
> > In the revised manuscript, we added more discussion in Section D.1 of Appendix due to the page limitation as follows.
> >
> > “We made two observations. First, RKD-MLP consistently performs better than the corresponding teachers over two perturbation scenarios. Specifically, while the performance of RKD-MLP and GNN teachers drops when the mask ratio increases, RKD-MLP always performs better than GNN teachers and other two baselines: MLP and GLNN. Second, our methods are more robust in topology noise than feature noise. For example, the performance gap between RKD-MLP and the second best baseline on incomplete graph structure (See figure 4) is higher than that on noise feature (See Figure 5). The possible reason is that when our model is well-trained, it only utilizes node features as input for prediction, leading to inevitable noises on the input data. In contrast, the node features are clean under the topology perturbation. These results demonstrate the effectiveness of the proposed meta-policy since it can filter out noisy teacher guidance.”
> >
> > **Q5.It would be interesting to discuss or compare the inference speed between pruning, quantification, and your method.**
> >
> > Thank you for the suggestion. In RQ6 of Section 4.2, we compared the inference speed of our method and pruning- and quantification- based SAGE model on two large-scale OGB datasets. The results are summarized in Table 3 of the revised manuscript.  We would like to stress that our method can bring more than 100x speedup in inference by removing the expensive aggregation in graphs (see Table 3 in the revised version). However, pure pruning and quantification cannot improve the speedup significantly. As shown in Table 3, the time cost of pruning and qualification based GNNs are almost the same. This is because the neighborhood aggregation operators of GNNs is the most time-consuming part, but pruning and qualification cannot address this issue. Since pruning and quantification is orthogonal to our learned MLP, in the future, we will investigate whether pruning and quantification can be used in our method to further improve the speed.
> >
> > **Q6.There are some typos, for example in the related work section “However,s it only”.**
> >
> > Thank you for pointing it out. We have fixed it. Please let us know if you see any other typos.

---

> > > ### Comment · Reviewer_asz6 · 2022-11-29
> > > **reply acknowledged**
> > >
> > > Thanks for the clarification. I will keep the score.

---

### Official Review · Reviewer_ZL6r · 2022-10-25

**Confidence:** 5
**Correctness:** 2
**Technical Novelty And Significance:** 1
**Empirical Novelty And Significance:** 2
**Recommendation:** 3

**Clarity, Quality, Novelty And Reproducibility:**

This paper has poor clarity and does not have sufficient quality and novelty. The code is released.

**Strength And Weaknesses:**

Strengths:

(+) How to learn a better MLP from GNNs is an interesting question. Considering GNNs might suffer from the scalability issue, how to learn an MLP as a replacement deserver some effort.

(+) This paper conduct experiments on nine datasets. In addition, the authors report mean and standard deviation for most of the experiments, which is nice.

Weaknesses:

(-) Vague and confusing motivation. The authors claim that “directly leveraging soft labels from the GNN teacher is suboptimal” but don’t have a reasonable explanation. Why “a large portion of unlabeled nodes will be incorrectly predicted by GNNs”? What “limited generalization ability”? The “100% accuracy on the training set, yet their test accuracy is merely around 80%” seems like an over-fitting problem. How is this performance gap between training and testing relevant to “leveraging soft labels from the GNN teacher is suboptimal”?

(-) Lack of novelty. The authors simply introduce a meta-policy to filter out unreliable soft labels based on node representations. The idea of distilling confident labels is not new [1, 2, 3], and methods using reinforcement learning meta-policy to determine confidence also exist [4, 5]. The author needs to reconsider the novelty of the proposed method and make new contributions, not just a combination of existing techniques.

[1] Müller, Rafael, Simon Kornblith, and Geoffrey E. Hinton. "When does label smoothing help?." NeurIPS, 2019

[2] Li, Yuncheng, et al. "Learning from noisy labels with distillation." ICCV. 2017

[3] Yang, Cheng, Jiawei Liu, and Chuan Shi. "Extract the knowledge of graph neural networks and go beyond it: An effective knowledge distillation framework." WWW, 2021

[4] Kostas, James, et al. "High Confidence Generalization for Reinforcement Learning." ICML, 2021

[5] Jordan, Scott, et al. "Evaluating the performance of reinforcement learning algorithms." ICML, 2020

(-) some technical details are unclear and incorrect. For example, Figure 1 (middle) is really confusing. GLNN contains an MLP student as the yellow line, what is the meaning of the red line MLPs-student? Are they different or are they the same? How is the MLPs-student implemented? Why the MLPs-student can perform much better (~0.78 accuracy) than the SAGE teacher when the noise ratio is zero? In the leaderboard of ogb-arxiv (https://ogb.stanford.edu/docs/leader_nodeprop/#ogbn-arxiv), the best model to date can only achieve 0.7719 accuracy, while MLPs-student already outperforms the state-of-the-art. I could not find any supporting claims/experimental results in the introduction or in the later experiment sections. The only time Figure 1 (middle) gets mentioned is in the preliminary section without any experimental details. I highly doubt the correctness of the figure. In addition, for the statement “GLNN-label is a variant of GLNN by excluding unlabeled nodes” - should it be GLNN-without-unlabel? All three models in Figure 1 (left) utilized labels, and GLNN-label cannot clearly distinguish the model.

(-) More supporting details are needed for some experiments. For example, In RQ2, the authors claim that “unlike small datasets, it is hard to train the MLP student on large graphs due to soft-label noises”. However, the soft-label noise difference between small and large datasets are not shown. In addition, what are the performances of SW, Entropy, and Cluster methods on large datasets?

(-) Lack of theoretical explanations or analyses. The motivation of this paper and the designs of the model is based on the “noisy soft labels”. However, why the accounting of the noises helpful is overlooked? How is the proposed model capture the noises and eliminate them? Why does elimination work? And how it works? How helpful can it be? For example, for Figure 4, the authors need to provide explanations for why RKD-MLP can perform well under noisy graph topology and noisy features. How are the graph topology and features relevant to the noisy soft labels?

(-) Poor readability. A lot of figures and claims in this paper are ambiguous and hard to understand, as I mentioned in the previous comments. For example, Figure 6 is really confusing. What do different colors indicate? What is the difference between different datasets? What is the meaning of the proposed model and GNN having the same predictions? In Figure 2, what is the meaning of different colors?

(-) Copy and paste Table 5 in RQ6 from the GLNN paper without reference. Table 5 in the paper is exactly the same as Table 4 in the GLNN paper. No reference is given. It seems the proposed method will not introduce any computational requirements in the inference stage. Will it introduce a heavy computation burden in the training stage?

[6] Zhang, Shichang, et al. "Graph-less neural networks: Teaching old mlps new tricks via distillation." ICLR, 2022.

**Summary Of The Paper:**

This paper proposes to distill a better MLP from the GNNs via reliable knowledge distillation. In particular, the authors introduce a meta-policy to filter out those unreliable soft labels. The authors conduct extensive experiments on multiple graph datasets.

**Summary Of The Review:**

The current version of the draft needs some work. The motivation is unclear and the paper lacks novelty in terms of its approach. The paper also needs to 1) double-check the correctness of some results and statements, 2) provide supporting details for the experiments, 3) provide explanations and analyses, and 4) improve the readability. Many parts of the paper are ambiguous and missing important details.

---

> ### Author Response · Authors · 2022-11-18
> **Response to Comments from Reviewer ZL6r (Part 1)**
>
> Dear reviewer,
>
> Thank you for the comments and feedback! Please see the response to your questions below.
>
> **Q1. Why “a large portion of unlabeled nodes will be incorrectly predicted by GNNs”?  What “limited generalization ability”? How is this performance gap between training and testing relevant to “leveraging soft labels from the GNN teacher is suboptimal”?**
>
> **The limited training samples lead to the poor generalization of GNNs on unlabeled nodes.** Given a graph $\mathcal{G}={\mathcal{V}, \mathcal{E}, \mathbf{X}}$ with label matrix $\mathbf{Y}$, in semi-supervised learning, we only have a small portion of nodes being labeled $(\mathcal{V}^L,\mathbf{Y}^L)$ and a large portion of nodes are unlabeled $(\mathcal{V}^U,\mathbf{Y}^U)$, i.e., $|\mathcal{V}^L|\ll |\mathcal{V}^U|$. If we train a GNN model based on the labeled nodes using $\mathcal{G}$, $\mathbf{X}$, and $\mathbf{Y}^L$, the trained GNN would naturally be impacted since deep models are label hungry to achieve good results. Consequently, when we apply the trained GNN to make predictions for unlabeled nodes in $\mathcal{V}^U$, a large portion of nodes in $\mathcal{V}^U$ will be wrongly classified, such as more than 20\% of nodes on Planetoid and OGB benchmarks. This phenomenon has been widely verified in state-of-the-art GNN methods.
>
> To distill a trained GNN model to an MLP student, the standard solution in the semi-supervised setting (as done in GLNN) is to use the soft-labels of both labeled and unlabeled nodes ($\{\hat{\mathbf{Y}}^L,\hat{\mathbf{Y}}^U\}$) as ground truth to train the MLP (minimize Eq.2). However, as mentioned above, the soft-labels of some unlabeled nodes are wrongly predicted, so the soft-label set $\{\hat{\mathbf{Y}}^L,\hat{\mathbf{Y}}^U\}$ contains a lot of noisy guidance. These noisy soft-labels would damage the optimization of MLP, leading to suboptimal performance of GLNN. It is worth mentioning that the test set in GLNN equals the unlabeled set $\mathcal{V}^U$.
>
> **Q2. The idea of distilling confident labels is not new [1, 2, 3], and methods using reinforcement learning meta-policy to determine confidence also exist [4, 5]. The author needs to reconsider the novelty of the proposed method and make new contributions, not just a combination of existing techniques.**
>
> We believe the reviewer has some misunderstandings of how our algorithm works. To address your concern, we compare them separately as follows.
>
> **Comparison between our model and works in [1,2,3].** In [1], the authors focus on label smoothing to improve the performance of the teacher model by commuting cross entropy with a weighted mixture of “hard” targets with the uniform distribution, i.e., $\forall c\in\mathcal{C} y_c^{LS}=y_c(1-\alpha) + \alpha/K$. As pointed out in Section 4 of [1], training the teacher network using label smoothing can lead to an inferior student model. Thus, label smoothing discussed in paper is not designed for distillation. In [2], as mentioned in the third paragraph of the Introduction section, [2] studies how to distill the knowledge learned from a small clean dataset to facilitate learning a better model from the entire noisy dataset. They do not focus on transferring knowledge from a better teacher model to guide the learning of a light but inferior model. Moreover, [2]r depends on a well-established knowledge graph (on the label space) to guide the distillation from noisy labels. It is nontrivial to obtain this precious knowledge graph. In contrast, we don’t require this side information. In [3], the authors do not consider noises in the teacher's prediction nor even mention this noisy issue. They adopt the same distillation objective as done in GLNN, and their main contribution is the design of a label propagation-based student model. However, we are working on designing a tailored knowledge distillation technique to avoid the impact of incorrectly predicted nodes on the learning of arbitrary student models.  To summarize, references [1] and [3] are not designed to distill knowledge from noisy teachers, and reference [2] requires a well-established knowledge graph describing relations between different labels to alleviate the noisy knowledge from the teacher.
>
> **Comparison between our model and works in [4,5].** In [4], confidence refers to the safety guarantees of reinforcement learning agents, which is confidence of reinforcement. Whereas, we use reinforcement learning to predict confidence, which is a completely different objective. The paper [5] focuses on how to evaluate reinforcement learning. We are unclear why the reviewer thinks this paper uses reinforcement learning to determine the confidence of soft labels from the teacher model. We are happy to address further comments from the reviewer regarding this. Using reinforcement learning to determine the confidence of the labels is a novel idea.

---

> > ### Author Response · Authors · 2022-11-18
> > **Response to Comments from Reviewer ZL6r (Part 2)**
> >
> > Sorry for splitting the responses into two parts because of the limits on the character number. Please read the left responses as follows:
> >
> > **Q3. Some technical details are unclear and incorrect. For example, Figure 1 (middle) is really confusing. Are they different or are they the same? How is the MLPs-student implemented? Why the MLPs-student can perform much better (~0.78 accuracy) than the SAGE teacher when the noise ratio is zero? In the leaderboard of ogb-arxiv (https://ogb.stanford.edu/docs/leader_nodeprop/#ogbn-arxiv), the best model to date can only achieve 0.7719 accuracy, while MLPs-student already outperforms the state-of-the-art. I could not find any supporting claims/experimental results in the introduction or in the later experiment sections. In addition, for the statement “GLNN-label is a variant of GLNN by excluding unlabeled nodes” - should it be GLNN-without-unlabel? All three models in Figure 1 (left) utilized labels, and GLNN-label cannot clearly distinguish the model.**
> >
> > Sorry for the confusion. In the revised manuscript, we have updated the name of “MLPs-student” in Figure 1(middle) to “GLNN-ideal” and illustrated the details in Section 2.2 (highlighted in blue font). Specifically, GLNN-idea follows the same setting of GLNN to set up the MLP student (i.e., 3-layer MLPs with dimension 256) and then uses the soft labels of correctly predicted nodes and a portion of wrongly predicted nodes via random sample to train the MLP student. It is worthy mentioning that GLNN utilizes the soft labels of correctly predicted nodes and all wrongly predicted unlabeled nodes to train the MLP student. However, GLNN-ideal can manually control the ratio of those wrongly classified nodes to train the same MLP student from the oracle perspective.
> >
> > Following this understanding, since the number of correctly classified nodes (including both labeled and unlabeled nodes) from the SAGE teacher (71.6%) is larger than the original training samples (53.7%) in ogbn-arxiv dataset, that is why GLNN-ideal with 0\% noises can perform better than SAGE teacher and other best models on the leaderboard. GLNN-ideal utilizes more training samples than the SAGE teacher (53.7% labeled nodes).
> >
> > Following your suggestions, we renamed “GLNN-label” to “GLNN-w/o-unlabel” in Figure 1(left).
> >
> >
> > **Q4. More supporting details are needed for some experiments. For example, In RQ2, the authors claim that “unlike small datasets, it is hard to train the MLP student on large graphs due to soft-label noises”. However, the soft-label noise difference between small and large datasets are not shown. In addition, what are the performances of SW, Entropy, and Cluster methods on large datasets?**
> >
> > This claim is made by the performance gap of GLNN on small and ogb datasets. For example, GLNN could achieve very competitive results with the SAGE teacher on three small Planetoid datasets (Cora, CiteSeer, and PubMed), but it loses to the SAGE teacher on ogbn-arxiv and ogbn-product datasets with a large margin. Specifically, as shown in Table 1 and Figure 3, the noise ratio on the typical small dataset (Cora) is 20.30% (1-79.70%), while 28.34% (1-71.66%) in ogbn-arixiv.
> >
> > Following your suggestions, we added additional experiments to test the performance of SW, Entropy, and Cluster on OGB datasets. The results are listed below.
> > |         | ogbn-arxiv | ogbn-products |
> > |---------|------------|---------------|
> > | SW      | 50.46      | 55.83         |
> > | Entropy | 61.57      | 65.62         |
> > | Cluster | 58.35      | 63.77         |
> >
> > We can observe that three heuristic methods do not perform well on the two large-scale and challenging OGB datasets. These results are added in Table 8 of the Appendix.

---

> > > ### Author Response · Authors · 2022-11-18
> > > **Response to Comments from Reviewer ZL6r (Part 3)**
> > >
> > > Sorry for splitting the responses into two parts because of the limits on the character number. Please read the left responses as follows:
> > >
> > > **Q5. Lack of theoretical explanations or analyses. The motivation of this paper and the designs of the model is based on the “noisy soft labels”. However, why the accounting of the noises helpful is overlooked? How is the proposed model capture the noises and eliminate them? Why does elimination work? And how it works? How helpful can it be? For example, for Figure 4, the authors need to provide explanations for why RKD-MLP can perform well under noisy graph topology and noisy features. How are the graph topology and features relevant to the noisy soft labels?**
> > >
> > > In this paper, we define “noisy soft labels'' as the soft labels of nodes that are wrongly classified by the GNN teacher. To explore the impact of those noisy nodes, in Section 2.2, we illustrated the impact of noisy soft labels on GLNN across various benchmark datasets in Figure 1(middle) and Figure 8 in the Appendix. The high-level idea is to manually control the ratio of noisy soft labels in the training of the MLP student, where the training set contains two parts: the soft labels of correctly classified nodes and the soft labels of wrongly classified nodes sampled.
> > >
> > > From Figure 1(middle) and Figure 8 in Appendix, we observe that “the soft labels of incorrectly predicted nodes restrict the capacity of MLPs students; By reducing the noise ratios, a stronger MLPs student can be easily achieved”. Motivated by this, we propose to train a meta-policy to automatically identify those wrongly predicted nodes to some extent, so that the MLP student can be learned upon fewer noises. In figure 6 of the updated version, we visualize the action made by our meta-policy. Basically, we observed that RKD-MLP can effectively detect nodes correctly predicted by the GNN teacher while filtering out nodes being wrongly classified to some extent (the gap between in group "0"). For instance, on the ogbn-arxiv dataset, RKD-MLP detects 93.34% (66.89/71.66) unlabeled nodes being correctly predicted by the teacher while filtering out 67.96% (19.26/28.34) nodes being wrongly predicted on the ogbn-arxiv dataset. This analysis is updated in RQ4 of Section 4.2.
> > >
> > > Following the same logic, since our method RKD-MLP can reduce the noise ratio to some extent, it can perform well on perturbation scenarios, i.e., topology perturbation and noisy features. Similarly, the correlation between noisy soft labels and graph topology and feature perturbations is:  when the input graph (topology or node features) of GNNs is corrupted, the performance of GNNs trained on the same training set is likely to drop, leading to decreased prediction results on unlabeled nodes and increasing the ratio of wrongly classified nodes.
> > >
> > > **Q6. Poor readability. A lot of figures and claims in this paper are ambiguous and hard to understand, as I mentioned in the previous comments. For example, Figure 6 is really confusing. What do different colors indicate? What is the difference between different datasets? What is the meaning of the proposed model and GNN having the same predictions? In Figure 2, what is the meaning of different colors?**
> > >
> > > Thank you for the detailed comments. Following your suggestions, we updated Figure 6 in the revised manuscript. Instead of reporting the confusion matrix, we visualize the meta-policy’s decisions w.r.t. the GNN teacher from two groups, where group 0 and 1 indicate the GNN teacher makes the right and wrong predictions, respectively. In light of this, we can clearly understand how many nodes being wrongly (or correctly) classified by the GNN teacher are filtered out (or preserved) by the proposed meta-policy.  For example, from the ogbn-arxiv result in Figure 6, we observe that RKD-MLP detects 93.34% (66.89/71.66) unlabeled nodes being correctly predicted by the teacher while filtering out 67.96% (19.26/28.34) nodes being wrongly predicted on the ogbn-arxiv dataset. This analysis is updated in RQ4 of Section 4.2.
> > >
> > > We updated Figure 2 in the revised version to have two node colors; one is for gold-labeled nodes, and the other is for soft-labeled nodes.

---

> > > > ### Author Response · Authors · 2022-11-18
> > > > **Response to Comments from Reviewer ZL6r (Part 4)**
> > > >
> > > > Sorry for splitting the responses into two parts because of the limits on the character number. Please read the left responses as follows:
> > > >
> > > > **7. Copy and paste Table 5 in RQ6 from the GLNN paper without reference. Table 5 in the paper is exactly the same as Table 4 in the GLNN paper. No reference is given. It seems the proposed method will not introduce any computational requirements in the inference stage. Will it introduce a heavy computation burden in the training stage?**
> > > >
> > > > Sorry for the confusion. In the revised manuscript, we moved the efficiency table to the main body as Table 3. Since our MLP student design follows exactly the same architecture of GLNN, the distilled student model has the same inference latency as that in GLNN. We clarified this similarity in the caption of Table 3.
> > > >
> > > > For the training stage, the additional computation cost is marginal due to two reasons: 1) the neural architectures of the student model and our policy network are simple MLPs, which are easy to optimize; 2) our meta-policy is a one-shot reinforcement learning, so the most tedious sampling process in standard reinforcement learning is avoided, making it faster to train. Based on our empirical results, the training cost of RKD-MLP is less than 1.8x that of GLNN.

---

### Decision · Program_Chairs · 2023-01-20

**Decision:**

Reject

**Justification For Why Not Higher Score:**

The main reason was the experimental issues (e.g. small or insignificant improvements, and issues with comparison with simple baselines as mentioned in meta-review), while the novelty issue was also a significant consideration.

**Justification For Why Not Lower Score:**

N/A

**Metareview: Summary, Strengths And Weaknesses:**

The paper proposes a distillation approach (RKD-MLP) to distill MLPs from GNNs, for the semi-supervised node classification task. Their approach uses reinforcement learning (RL) to filter out noisy soft labels from the teacher (GNN) model.

In general, reviewers find the paper to be clearly written, and well-motivated, and appreciate the simple and intuitive RL-based approach. However, there are a number of main issues raised by reviewers:

- Experimental comparisons: performance improvements of the proposed method over the Entropy and Cluster baselines is small in many cases (and seem to be not statistically significant in some cases). This makes its benefits less clear considering that the RL approach seems more complex with likely higher training time than these simple heuristics, and also requires significantly more hyperparameter tuning (with around 5 hyperparameters to tune, with up to 10 choices each); it is also unclear on what data the tuning was done (considering that the validation set is already used for the RL algorithm). Reviewers also note that it is unclear why the proposed method is mainly compared against GLNN instead of Entropy and Cluster (for calculating the percentage improvement) when GLNN has usually weaker performance than 1 or both of these. These baselines are also seem to be not considered in some of the other experiments, such as the large-scale graphs (Fig 3) and the added experiments on the CPF teacher.

- Technical novelty: some reviewers note that the paper can be seen as mainly combining some existing ideas of GLNN (knowledge distillation) and reinforcement learning. The author response points to the novelty of the RL formulation for filtering. They also state that the RL method avoids needing a hyperparameter for the threshold, though this is less convincing considering that it introduces a few other hyperparameters.

- Some other points were noted but satisfactorily addressed by the author response, including the use of validation set, SOTA teacher model, and improvements to paper presentation. I thank the authors for their efforts in addressing concerns and improving the paper.

In the end, reviewers and AC agree that while the work has promising merits, due to these issues, the work is not yet ready for publication at ICLR. The reviews offer a number of helpful suggestions for improvement, so I encourage the authors to continue improving the paper based on the reviews for future submissions.